# Actin polymerization promotes invagination of flat clathrin-coated lattices in mammalian cells by pushing at lattice edges

Changsong Yang[1], Patricia Colosi[2], Siewert Hugelier [2], Daniel Zabezhinsky[1], Melike Lakadamyali [2] & Tatyana Svitkina [1] ✉

Clathrin-mediated endocytosis (CME) requires energy input from actin polymerization in mechanically challenging conditions. The roles of actin in CME are poorly understood due to inadequate knowledge of actin organization at clathrin-coated structures (CCSs). Using platinum replica electron microscopy of mammalian cells, we show that Arp2/3 complex-dependent branched actin networks, which often emerge from microtubule tips, assemble along the CCS perimeter, lack interaction with the apical clathrin lattice, and have barbed ends oriented toward the CCS. This structure is hardly compatible with the widely held "apical pulling" model describing actin functions in CME. Arp2/3 complex inhibition or epsin knockout produce large flat non-dynamic CCSs, which split into invaginating subdomains upon recovery from Arp2/3 inhibition. Moreover, epsin localization to CCSs depends on Arp2/3 activity. We propose an "edge pushing" model for CME, wherein branched actin polymerization promotes severing and invagination of flat CCSs in an epsin-dependent manner by pushing at the CCS boundary, thus releasing forces opposing the intrinsic curvature of clathrin lattices.

Clathrin-mediated endocytosis (CME) is the major endocytic pathway in cells, which is involved in multiple physiological and pathological processes[1,2]. The formation of an endocytic vesicle during CME involves local membrane invagination to form a small bud followed by constriction of the bud neck and vesicle scission and departure. All these events are energetically unfavorable, but the relevant sources of energy are not fully understood.

Clathrin self-assembly is an important but often not sufficient force-generating mechanism for membrane invagination during CME. Energy input from actin polymerization is necessary for CME in yeast, and important in mammalian cells[2–5], especially when high membrane tension resists invagination[6–8]. In mammalian cells, actin is thought to function only at the late CME stages[7,9–11], whereas in budding yeast actin polymerization was found to promote the initial invagination of flat clathrin-coated structures (CCSs)[12]. Besides CME, large flat clathrin-coated plaques serve as unconventional cell-matrix adhesions and

mechanosensitive signaling platforms[6,13]. The turnover of these plaques requires actin dynamics[14], but the underlying mechanism is unclear.

Actin filaments function in CME in the form of branched networks generated by the Arp2/3 complex[15–18]. In general, branched actin networks produce pushing force by elongating multiple barbed ends against the load[19]. Accordingly, at a structural level actin filament barbed ends are oriented toward the target – the plasma membrane in lamellipodia[20,21] or a moving particle in comet tails[22]. Using platinum replica electron microscopy (PREM), we have found that barbed ends are also oriented toward invaginating CCSs in B16F1 melanoma cells[18]. Importantly, these branched actin networks were located at the CCS perimeter without associating with the apical CCS surface. Based on these finding, we proposed that barbed ends promote bud elongation, neck constriction and vesicle departure after scission by pushing onto the edge of the clathrin coat without forming any strong interactions

[1]Department of Biology, University of Pennsylvania, Philadelphia, PA, USA. [2]Department of Physiology, Perelman School of Medicine, University of Pennsylvania, Philadelphia, PA, USA. ✉e-mail: svitkina@sas.upenn.edu

with the clathrin coat itself[18] (Fig. 1a, left). Here, we refer to this model as "edge pushing" mechanism.

Our observations made in mammalian cells[18] appear inconsistent with the model proposed for CME in budding yeast[23–25], which postulates that while barbed ends of the branched network push onto the plasma membrane around the CCS base, the older parts of the network containing branch junctions are attached to the clathrin coat by molecular linkages, such as Sla2 (Hip1R in human) and Ent1 (epsins in human)[25], thus allowing the actin retrograde flow to pull the anchored CCS inward (Fig. 1a, right). This "apical pulling" model predicts that the barbed ends are oriented radially away from the center of the CCS[23], and that the invaginating CCSs should be extensively covered by stably anchored branched actin filaments. Direct visualization of actin organization at CCSs in yeast has not been achieved. Electron microscopy (EM) of thin-sectioned plastic-embedded yeast cells revealed a ribosome-free area around CCSs, which contained actin and actin-binding proteins typically associated with branched actin networks[12,26–28]. However, individual actin filaments in this zone were not resolved leaving open the question of how actin filaments are organized at endocytic sites. Despite the discrepancy between the predictions of the yeast model of CME[23–25] and direct PREM observations in mammalian cells[18], the apical pulling model is frequently extrapolated, implicitly or explicitly, to mammalian cells[3,23,24,29–32]. This trend motivated us to address functional contribution of Arp2/3 complex and epsins to morphogenesis of CCSs in multiple mammalian cells. We focused on roles of epsins rather than Hip1R because the Hip1R recruitment to CCSs in mammalian cells depends on epsins[33].

Here, using PREM[34,35] to simultaneously visualize the CCS shape and the architecture of CCS-associated actin networks, we show that branched actin polymerization promotes invagination of flat CCSs formed on the adherent plasma membranes, thus functioning at much earlier CME stages than previously thought. Moreover, we demonstrate that an important component of actin-assisted CCS invagination is an ability of branched actin networks to split large flat clathrin-coated plaques into smaller units that subsequently form normally sized clathrin-coated buds and vesicles and that this function critically requires the presence of epsin family proteins, whereas epsin localization to CCSs in turn depends on branched actin networks. Branched actin networks mediating both invagination and splitting of CCSs originate not only from preexisting unbranched actin filaments, as we reported previously[18], but strikingly, also from microtubule tips. At all stages of CCS morphogenesis, the branched actin networks are associated with the CCS perimeter and have barbed ends oriented toward the CCS. These data demonstrate that structural organization of CME sites in mammalian cells is incompatible with the apical puling model of actin roles in CME and, therefore, that the mechanisms by which the actin cytoskeleton promotes CME in yeast and mammalian cells could be significantly different.

## Results

Knowledge of actin filament organization in CCS-associated actin patches is absolutely necessary for full understanding of the mechanisms by which the actin polymerization promotes CCS dynamics. Fluorescence microscopy has been instrumental for revealing CCS dynamics in living cells[36–39], but even most advanced versions of superresolution fluorescence microscopy are unable to resolve individual actin filaments. This task is also challenging for EM due to dense packing of branched actin filaments, small sizes of CCS-associated actin patches, and a well-known difficulty of preserving dynamic actin filament networks for EM. PREM is the highest resolution method currently available to reveal structure of three-dimensional actin filament arrays at a single-filament level and over large cellular spaces[34,35]. In this study, we exposed the CCSs for PREM analyses in cultured mammalian cells by an "unroofing" procedure that

mechanically removes the cell top while preserving the ventral plasma membrane with associated structures[18,40].

## Morphological categorization of CCSs in different cell types

We investigated the distribution of CCS morphologies and their association with branched actin networks in four mammalian cells types: U2OS and HeLa cells, both endogenously expressing GFP-dynamin2 and RFP-clathrin light chain (CLC)[41], as well as PtK2 and B16F1 cells. We quantified frequencies of different CCS categories: flat (Fig. 1b, blue shade); dome-shaped (Fig. 1b, green shade), and spherical CCSs (Fig. 1b, orange shade). These data showed that U2OS (Fig. 1 and Supplementary Fig. 1), PtK2 (Supplementary Fig. 2) and HeLa (Supplementary Fig. 3) cells had substantial fractions of all three CCS shape categories, whereas B16F1 cells (Supplementary Fig. 4a–e) contained a negligible amount (-1%) of flat CCSs (Fig. 1e). The previously observed distribution of CCS shapes in HeLa cells[42] is consistent with our results. The projection area (Fig. 1f) and linear dimensions (Supplementary Fig. 3f, g) of spherical and dome-shaped CCSs were relatively uniform. In contrast, the size of flat CCSs varied broadly from as small as -5×$10^3$ nm$^2$, which corresponded to just a few clathrin polygons (-70 nm in diameter) (Fig. 1b, inset; Supplementary Fig. 3f, g), to as large as -200×$10^3$ nm$^2$ (up to 1 μm across).

Flat CCSs were remarkable not only because of their size, but also because of the presence of frequent disruptions of the polygonal clathrin pattern (Fig. 1c), as reported previously[43,44]. These disruptions often followed a linear path, thereby creating "seams", which divided the clathrin lattice into subdomains approximately of the size of a typical dome-shaped or spherical CCS. The subdomains could exhibit different curvatures within the same large CCS (Fig. 1d), suggesting that large flat CCSs can produce endocytic buds and vesicles of regular size, while simultaneously undergoing disassembly, as was observed previously by light microscopy[42,45,46].

## Relationship of branched actin networks and CCSs

Structural relationship between a CCS and the associated branched actin network is a key clue for revealing the mechanism, by which actin dynamics can promote CME. The main outstanding questions are whether the CCS invagination mechanism includes extensive interactions between actin filaments and the clathrin lattice and whether actin barbed ends are oriented toward or away from the clathrin lattice.

We found that a significant subset of CCSs in each morphological category was associated with branched actin networks (Fig. 1b, yellow shade) in all examined cell types (Fig. 1e). Importantly, in all cases, the branched actin networks localized adjacent to the CCS and were largely absent from most of its apical surface (Fig. 2 and Supplementary Figs. 1–4, Supplementary Movies 1–3). Branched actin networks often originated from long linear actin filaments in the background, with which they formed a single-point contact at their pointed-end side (Fig. 1b, arrows)[18]. These long filaments could serve as original mother filaments for initiation of branched nucleation, as well as provide traction allowing efficient pushing onto the CCS.

Branched actin networks that were associated with spherical CCSs often formed a comet tail-like arrangement (Fig. 2a, b, e; Supplementary Figs. 2d, h; 3b, c, and 4e, Supplementary Movies 1–3). In occasional side views, we could see that the branched actin network appeared to push onto uncoated (naked) membrane at the base of the spherical CCS rather than on the clathrin coat (Fig. 2b). Branched actin networks also could surround a spherical CCS like a collar with actin filament ends often reaching underneath the vesicle toward its neck while leaving most of the clathrin coat exposed (e.g. Figure 2c, d; Supplementary Figs. 2f; 4b, c, Supplementary movies 1–3). These structural arrangements are consistent with the idea that the Arp2/3 complex is activated by membrane-associated nucleation-promoting factors. Peripheral localization of branched actin networks, as well as orientation of actin filament ends toward the CCS, were also apparent in the

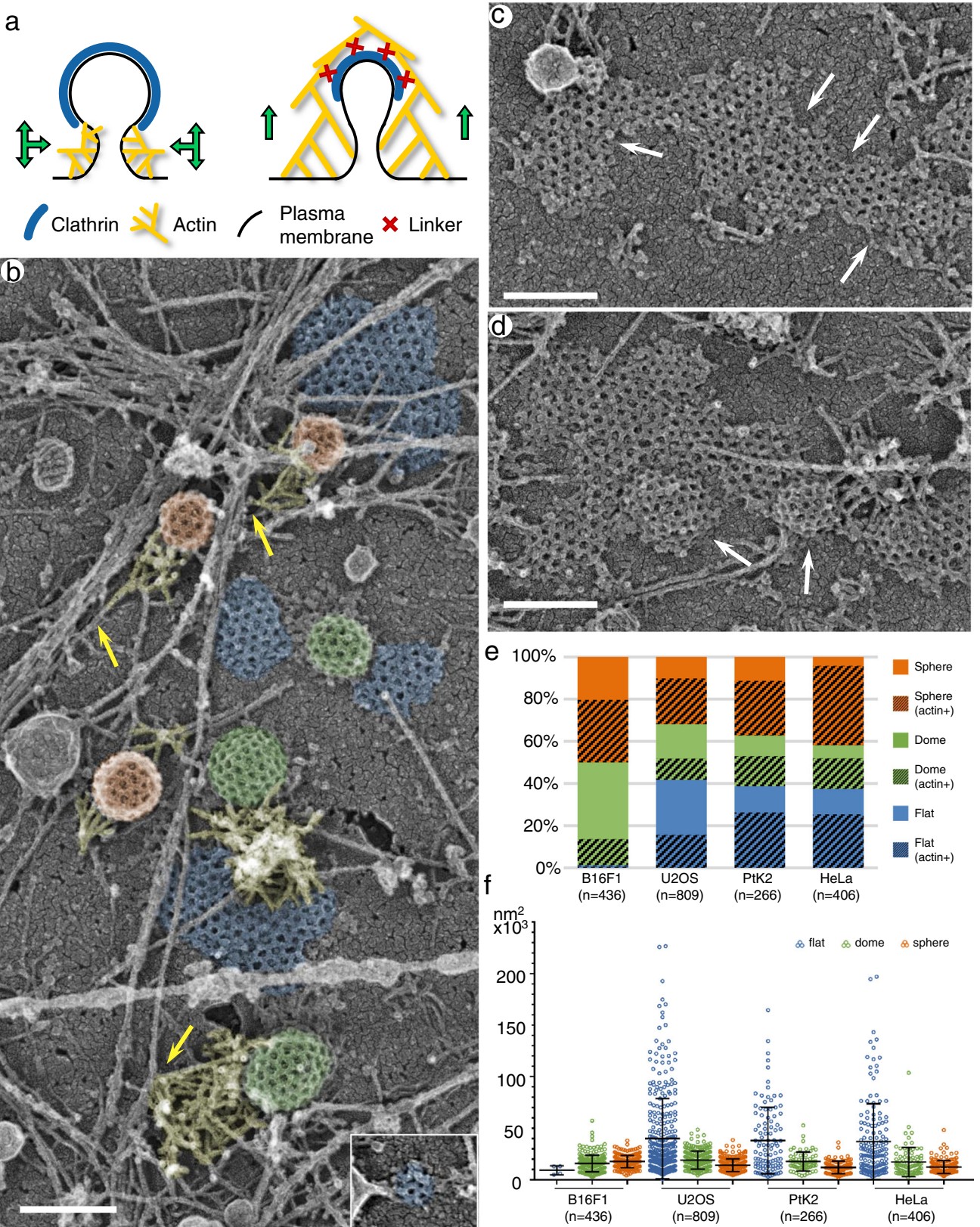

**Fig. 1 | PREM analysis of CCS morphology. a** Two alternative models for actin-dependent invagination of CCSs. (Left) Edge pushing model. (Right) Apical pulling model. **b** A region of an unroofed U2OS cell showing flat (blue), dome-shaped (green) and spherical (orange) CCSs, some of which associate with branched actin networks (yellow). Arrows mark linear actin filaments from which branched networks originate. Inset shows a very small flat CCS. **c** Large flat CCSs in U2OS cells contain subdomains demarcated by "seams" (arrows) with disrupted polygonal clathrin pattern. **d** Dome-shaped subdomains (arrows) within a large mostly flat CCS in U2OS cells. **e** Percentages of different CCS shape categories in four different cell types. Hatched colors indicate a fraction of actin-associated CCSs within a given category. **f** Projection area of CCSs of different shape categories in indicated cell types. Error bars, mean ± SD. Scale bars, 200 nm.

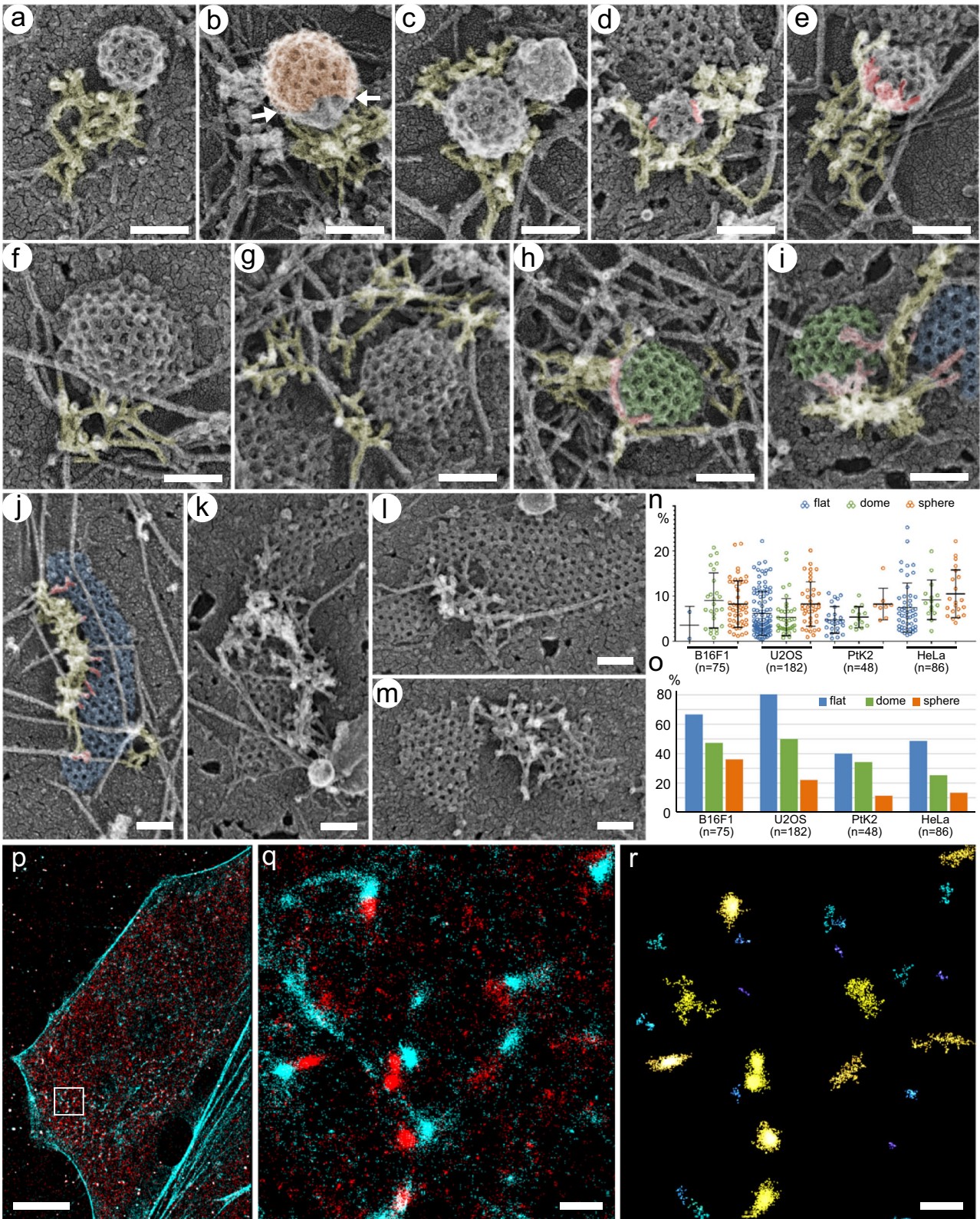

case of dome-shaped CCSs (Fig. 2f–i; Supplementary Figs. 2b, c; 3d; 4d).

Since we previously used B16F1 cells, which largely lack flat CCSs[18], it remained unclear whether actin filaments can function at flat CCS. Using other cell types, we found that branched actin networks, surprisingly, were also frequently associated with flat CCSs, even though the actin cytoskeleton was previously thought to function only at already invaginated CCSs[7,9–11]. At flat CCSs, branched actin networks were typically arranged along the CCS perimeter (Fig. 2j; Supplementary Figs. 2g; 3f) and sometimes entered into deep "bays" in the clathrin lattice (Fig. 2l; Supplementary Fig 3h). Branched actin networks could also be seen to cross a flat CCS in a linear manner, as if following a clathrin lattice seam (Fig. 2k; Supplementary Figs. 2g; 3e), or to reside between the neighboring CCSs of different shapes (Fig. 2m,

**Fig. 2 | Branched actin networks localize at the CCS perimeter with minimal overlaps with the clathrin lattice. a–e** Spherical CCSs in U2OS cells. Branched actin networks (yellow) can form a lateral comet tail (**a**, **b**, **e**), surround the CCS (**d**), or localize mainly underneath the CCS (**c**). **b** Branched actin network contacts an uncoated part (neck) of the CCS (clathrin coat is shown in orange). **d**, **e** Examples of actin filaments extending over the clathrin lattice (red); **e** shows a maximal observed level of actin-CCS overlap. **f–i** Dome-shaped CCSs in U2OS cells. Branched actin networks are laterally associated with CCSs (**f**, **g**) with occasional overhangs over the CCS periphery (**h** and **i**, red). **i** Branched actin network located in part at the interface between dome-shaped (green) and flat (blue) CCSs. **j–m** Flat CCSs in U2OS cells. Branched actin networks associate with the flat CCS (blue) margin (**j**), partially surround the CCS and separate it from the neighboring CCS (**k**), occupy a "bay" in

the CCS (**l**), or localize between two neighboring CCSs (**m**), in all cases exhibiting minimal overhangs of actin filament ends over the CCS (red in **j**). **n** Percentage of the CCS area overlaid by branched actin filaments in 2D projection images within the actin-positive CCS subpopulations in the indicated cell types. Error bars, mean ± SD. **o** Fraction of the actin-positive CCSs with any amount of branched actin extending over their lattice in 2D projection images in the indicated cell types. **p–r** STORM imaging of intact fixed PtK2 cells stained with AlexaFluor488-phalloidin (cyan) and AlexaFluor405-647-labeled antibody against CHC (red). **p** Cell overview. **q** Zoomed region outlined by white box in **p. r** Region in **q** after Voronoi segmentation of clathrin localizations. Scale bars: 100 nm (**a–m**), 10 μm (**p**) and 500 nm (**q**, **r**).

Supplementary Figs. 2c, d; 3g). In these cases, actin filament ends were oriented toward each of the neighbors, as if pushing them apart. In virtually no cases we observed actin filament branch junctions, where the pointed ends are captured by the Arp2/3 complex, to preferentially overlap with the clathrin lattice while having their barbed ends oriented in the centrifugal direction.

We observed only minimal overlaps of the branched actin network and the clathrin lattice. We argue that a lack of actin interaction with the apical CCS surface is not a consequence of mechanical disruption by unroofing. To avoid such a concern, for our PREM analyses, we selected samples that retained a high degree of three-dimensionality, in which actin networks typically extended upward well above the height of CCSs. Yet, these networks encircled the CCSs and did not interact with the interior clathrin lattice. If such arrangements were a result of actin network rupture, then one needs to assume that the unroofing procedure drills channels in the actin network exactly above the clathrin lattice, while leaving the networks in immediate vicinity intact, which is clearly an unrealistic suggestion. Moreover, if unroofing procedure detached actin from the top of a CCS, it would also shave away other cellular components in the same Z-dimension. However, other structures that accidentally crossed a CCS often remained preserved on the apical CCS surface. Thus, peripheral localization of branched actin networks at CCSs cannot be explained by a loss of apical actin networks due to mechanical impact.

The cases of actin-clathrin overlaps, when present, were typically restricted to the CCS edge (Fig. 2a–m; Supplementary Figs. 1-4). The average percentage of the CCS projection area overlaid by actin filaments across all four cell types and all three CCS morphological categories was as low as 7.1 ± 5.0% (mean ± SD, $n = 391$ actin-positive CCSs) (Fig. 2n), suggesting that physical interaction between the clathrin coat and the branched actin network cannot be strong. The occasional relatively high values (still below 30%) in this data set (Fig. 2n) were rare and resulted from either a small size of the CCS (Supplementary Fig. 2c), or a side view of spherical CCS pushed by a comet tail (Fig. 2e), or an unusually large size of the branched actin network (e.g. Figure 2k, m; Supplementary Fig. 3f). Importantly, even in these cases, orientation of free filament ends (presumably, barbed ends) was toward, but not away, from the CCS center (e.g. red regions in Fig. 2d, e, h, i, j). Further support for a lack of robust interaction between actin filaments and the clathrin lattice was provided by samples produced by relatively stringent unroofing. In these cases, most cytoskeletal components were removed, but the remaining bits of actin filaments were attached to the CCS perimeter and not to the apical surface of the clathrin lattice (Supplementary Fig. 2e). These findings suggest that the plasma membrane just outside of the CCS represents the strongest attachment site for actin filaments, whereas anchorage of the network to the lattice itself is weak at best, but more likely is absent altogether.

The actin-CCS overlaps were not only miniscule in their area (Fig. 2n), but also observed only in a subset of actin-associated CCSs (Fig. 2o). Importantly, the actin-CCS overlaps were more common for flat CCSs, but their frequencies decreased progressively as the CCS

curvature increased from flat to dome-shaped and then from dome-shaped to spherical CCSs (Fig. 2o). If physical interaction between the clathrin coat and the branched actin network was essential for CCS invagination, we would expect an opposite trend – more branched actin would be associated with the clathrin lattice of CCSs undergoing active invagination. Therefore, the observed overlaps are more likely explained by an exuberant multidirectional nucleation of daughter filaments in branched actin networks, which find less obstruction at the edges of flat CCSs as compared with dome-shaped CCSs, which in turn present less hindrance than spherical CCSs. Moreover, actin filaments that overlapped with more central parts of the clathrin lattice in 2D projections exhibited no apparent physical interaction with the apical clathrin coat, if viewed in 3D using tilted PREM images (Supplementary Movies 1-3), further arguing against strong interaction between the coat and branched actin filaments.

If CME in mammalian cells was promoted through extensive interactions of actin filaments with the entire clathrin lattice, as proposed by the apical pulling model, most CCSs would be concealed underneath branched actin filaments in PREM samples, in which case we might miss them. To address this possibility, we examined all discrete patches of branched actin networks (excluding large peripheral networks that likely functioned in membrane protrusion) in our PREM samples. We found that vast majority of such patches (74.8–88.9% depending on the cell type) were associated with well-visible CCSs or other vesicles, i.e. caveolae or uncoated vesicles (Supplementary Fig. 4h). The fraction of actin patches, which were not already associated with vesicles and had an adequate area, density and geometry (e.g. not too narrow) to conceal a CCS with a diameter of 125 nm, comprised from 4.4% of all patches in U2OS cells to 9.8% in B16F1 cells, with intermediate values for PtK2 (4.8%) and HeLa (6.6%) cells (Supplementary Fig. 4h). However, even these actin patches typically were rather flat with an insufficient height to conceal a dome-shaped or spherical CCS. The 125 nm diameter is a value slightly below an average diameter of dome-shaped and spherical CCSs in our datasets (Supplementary Fig. 4f, g), which we chose to be on a conservative side. These data indicate that a fraction of CCSs potentially hidden underneath branched actin networks is negligible compared with CCSs that obviously interact with branched actin networks only at their perimeter.

To further address a minor possibility of structural damage, we performed superresolution fluorescence microscopy of phalloidin-stained F-actin and immunolabeled clathrin in intact PtK2 cells by two-color stochastic optical reconstruction microscopy (STORM) (Fig. 2p–r and Supplementary Fig. 5). We analyzed STORM data using quantitative cluster analysis to define actin and clathrin clusters and then performed actin-clathrin colocalization analysis within local region surrounding each clathrin cluster (see Methods and Supplementary Fig. 5a). This analysis showed that only 22.15 ± 14.25% of F-actin localizations (mean ± SD, $n = 5752$ clathrin clusters from 10 cells in two independent experiments) colocalized with clathrin clusters (Supplementary Fig. 5d), whereas an average density of actin localizations was significantly lower within the outlines of clathrin clusters

than within actin clusters located immediately outside of clathrin clusters (Supplementary Fig. 5e). Earlier observations by diffraction-limited fluorescence microscopy[11] and more recent data by super-resolution microscopy[31] also showed that F-actin was spatially off-center of the associated CCSs. The degree of the F-actin/clathrin overlap in our STORM images is likely overestimated, because some of colocalized clathrin-F-actin localizations can represent actin filaments that push onto the neck of Ω-shaped CCSs or unbranched actin fila-ments that are present throughout the cell because phalloidin staining does not distinguish branched and unbranched actin filaments. Even with these limitations, the observed low level of overlap is incompa-tible with the idea that CCS invagination heavily relies on the F-actin interaction with the clathrin lattice.

Overall, these data show that structural organization of branched actin networks relative to CCSs in mammalian cells is inconsistent with the apical-pulling model proposed for yeast CME, in which actin fila-ment barbed ends are predicted to face away from the CCS and older parts of the network containing pointed ends and branch junctions to interact with the clathrin coat and be present on its apical surface.

### Arp2/3 inhibition blocks CCS invagination and size control

The actin cytoskeleton in budding yeast stimulates invagination of newly formed flat CCSs[2,12], whereas in mammalian cells actin dynamics is thought to function only after the initial invagination has been accomplished by other mechanisms[7,9–11]. However, the frequent asso-ciation of branched actin networks with flat CCSs in our PREM samples suggested that actin polymerization can also promote invagination of flat CCSs in mammalian cells. To test this idea, we investigated the effects of CK-666, a specific inhibitor of the Arp2/3 complex, on CCS morphogenesis. Treatment was performed in a low-serum (0.1% FBS) medium for efficient inhibition of the Arp2/3 complex. When cells were treated with CK-666 in the presence of 2-10% FBS, they still retained abundant uninhibited branched actin networks that can be seen by PREM. Although some branched networks were still detectable even at 0.1% FBS, the efficiency of Arp2/3 complex inhibition by CK-666 was much improved.

PREM analysis of CK-666-treated PtK2 (Fig. 3, Supplementary Movie 4), U2OS (Supplementary Fig. 6) and HeLa (Supplementary Fig. 7) cells demonstrated that Arp2/3 complex inhibition led to severe depletion of invaginated (dome-shaped and spherical) CCSs, accompanied by a dramatic increase in the frequency and size of flat CCSs (Fig. 3g, h; Supplementary Figs. 6f, g; 7f, g). As compared with cells cultured in the presence of 10% FBS, treatment with 0.1% FBS and DMSO also increased the frequency and size of flat CCSs (likely because low-serum conditions dampen Arp2/3 complex activation), although to a much lesser degree than CK-666 treatment (Fig. 3g, h). Importantly, CCSs in cells incubated with 0.1% FBS remained extensively associated with branched actin networks (Fig. 3a–d, g; Supplementary Figs. 6a; 7a, b, Supplemen-tary Movie 5), whereas majority of CCSs in CK-666-treated samples lacked associated branched actin networks. Long unbranched actin filaments were seen throughout the sample, with and without CK-666 treatment, and could occasionally be found in the CCS vicinity or even cross the clathrin lattice (Fig. 3e, f; Supplementary Figs. 6b, c; 7c, d, Supplementary Movie 4), but they did not appear to have a functional relationship with CCSs. Uninhibited branched actin networks only infrequently were found in CK-666-treated samples, and they were usually rather small (Fig. 3f; Supplementary Figs. 6b, c; 7c, Supplementary Movie 4). The incomplete inhibition of Arp2/3 complex activity could be responsible for a small proportion of remaining dome-shaped and spherical CCSs (e.g. Figure 3f, Sup-plementary Fig. 6c). The CK-666 treatment even induced numerous flat CCSs in B16F1 cells, which virtually lacked flat CCSs in normal conditions (Supplementary Fig. 8), even though Arp2/3 complex was only partially inhibited in this case, because we used milder

treatment conditions (10% FBS, 100 μM CK-666 for 1 h) to avoid cell detachment in this sensitive cell type.

Consistent with the known adhesive functions of clathrin plaques in some cell types[6,13], the flat CCSs in CK-666-treated U2OS and HeLa cells appeared to serve as cell adhesion sites. Indeed, after strong unroofing when most of the cellular components were removed by sonication, the remaining plasma membrane fragments almost invariably contained, or even were fully occupied by flat CCSs (Sup-plementary Figs. 6d; 7e). We also noticed some peculiar rod-shaped structures in CK-666-treated samples in all three tested cell types (PtK2, U2OS and HeLa), which appeared as bundles of "beaded" strands (Supplementary Fig. 6e). We never detected such structures in untreated cells. They likely represent actin-cofilin rods, which are formed in cells under stressful conditions or when monomeric actin is present in excess[47].

### Inhibition of the Arp2/3 complex impairs CCS dynamics

We next sought to evaluate the effects of Arp2/3 complex inhibition using TIRF microscopy of intact cells (Fig. 4). Treatment with CK-666 led to the formation of large clathrin-positive structures instead of typical small scattered dots (Fig. 4a–c). Quantification of TIRF micro-scopy images showed a significant increase in the CCS area in all three examined cell types (Fig. 4d), consistent with our PREM data, but shown for intact cells.

To evaluate CCS dynamics after downregulation of Arp2/3 complex activity, we performed live cell imaging by TIRF microscopy of U2OS and HeLa cells endogenously expressing RFP-CLC and treated with DMSO or CK-666 in low serum conditions (Fig. 4e–l, Supplementary Movie 6). By measuring CCS lifetimes in kymographs generated from 10-min long sequences (Fig. 4e–h), we found that an average CCS lifetime per cell was dramatically increased after CK-666 treatment for both U2OS (Fig. 4i) and HeLa (Fig. 4k) cells. Fur-thermore, in kymographs of control DMSO-treated cells, most CCS traces were shorter than the length of the movie (600 sec) indicating dynamic appearance and/or disappearance of CCSs. In contrast, after CK-666 treatment, a large fraction of CCSs was present throughout the movie (Fig. 4j, l). Thus, inhibition of the Arp2/3 complex not only increases the size of CCSs, but also severely stalls their dynamics.

Together, our experiments addressing the roles of the Arp2/3 complex in CME show that assembly of Arp2/3-dependent branched actin networks is required for CCS invagination and dynamics in mammalian cells. Moreover, the accumulation of very large flat CCSs after inhibition of the Arp2/3 complex suggests that branched net-works also regulate the proper size of CCSs to make them suitable for CME.

### Restored actin networks drive CCS division and invagination

The pharmacological approach uniquely allows monitoring the time course of recovery after removal of the inhibitor with high temporal resolution. In our case, we washed out CK-666 to determine how newly restored branched actin networks reverse the phenotype of Arp2/3 complex inhibition. The cells were then unroofed with ~30 s intervals and examined by PREM (Figs. 5 and 6, Supplementary Movies 7-10).

The CCS-associated branched actin networks reappeared already after 30 s of CK-666 washout and increased over time in size and fre-quency (Fig. 5a). In parallel, the fraction and size of flat CCSs pro-gressively decreased, while fractions of dome-shaped and spherical CCSs increased, until by the 2 min time point the distribution of CCS morphologies was roughly similar to that of untreated cells (Fig. 5a, b).

The patches of branched actin networks formed after 0.5-min washout were relatively small and located at the edges of flat CCSs (Fig. 5c, d, pink arrowheads; Supplementary Movie 7). Interestingly, they were also found in the bays of the clathrin lattice and along the

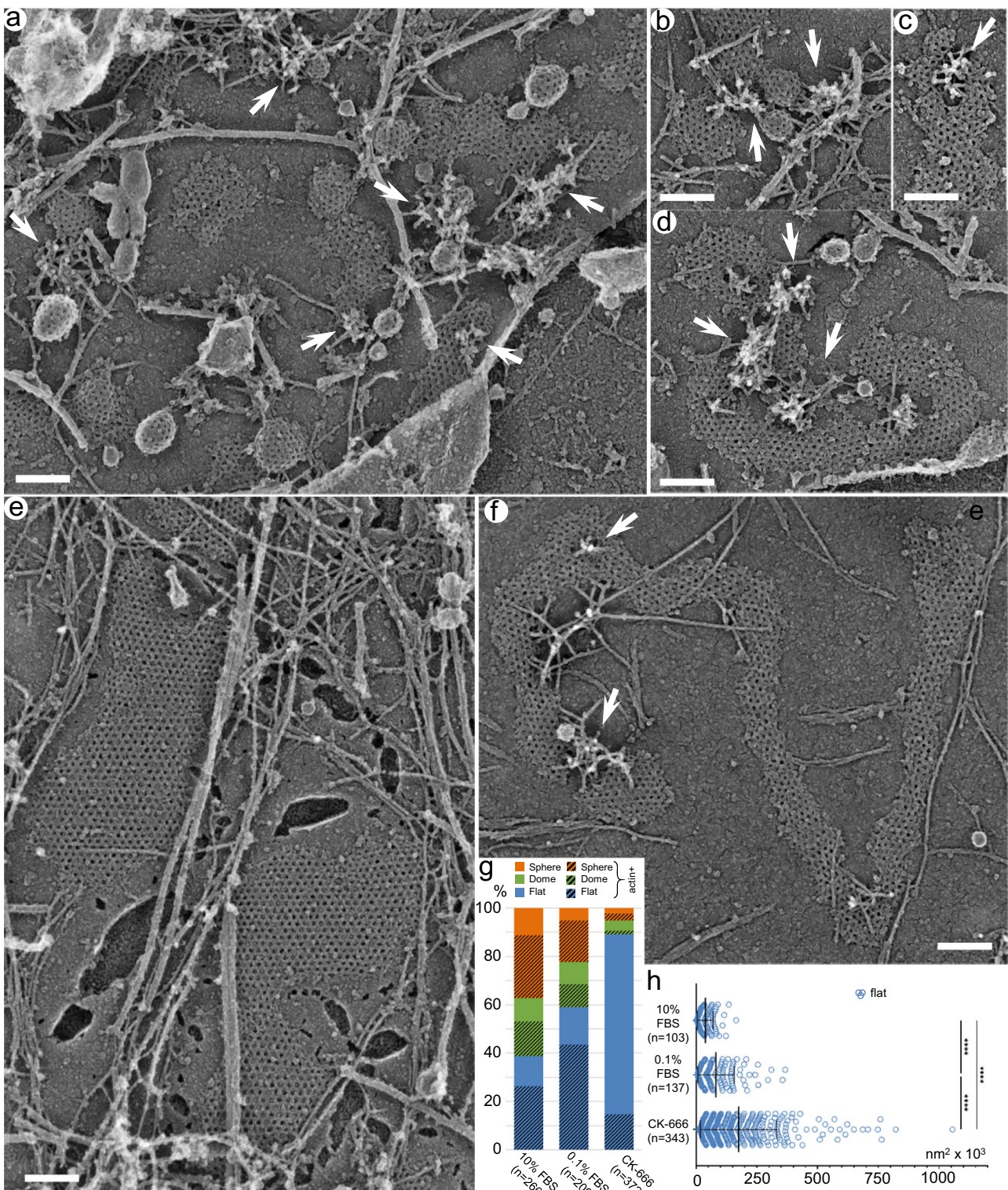

**Fig. 3 | Inhibition of the Arp2/3 complex results in accumulation of large flat CCSs at the expense of other CCS categories. a–d** Control PtK2 cells cultured in the presence of 0.1% FBS and DMSO exhibit flat, dome-shaped and spherical CCSs, some of which are associated with branched actin networks (arrows). **e, f** PtK2 cells treated with 200 μM CK-666 in the presence of 0.1% FBS contain very large flat CCSs occasionally associated with small remaining patches of branched actin network (arrow). Scale bars, 200 nm. **g** Percentage of different CCS shape categories in PtK2 cells in indicated conditions. Hatched colors indicate a fraction of CCSs associated with branched actin networks within the given category. Data for 10% FBS are the same as in Fig. 1. **h** Projection area of flat CCSs in PtK2 cells in indicated conditions. Error bars, mean ± SD; ****$p < 0.0001$ (Kruskal-Wallis with posthoc Dunn's multiple comparison test).

lattice seams (Fig. 5c–j, yellow arrowheads), suggesting that branched actin networks might create these bays and were actively involved in the division of flat CCSs by invading into CCS seams and pushing them apart. The decrease in the area of flat CCSs that was observed already at the 0.5 min time point most likely reflected the division of preexisting flat CCSs. Actin assembly along CCSs margins was clearly increased after 1 min of CK-666 washout, when small patches of branched actin network converted into elongated strands

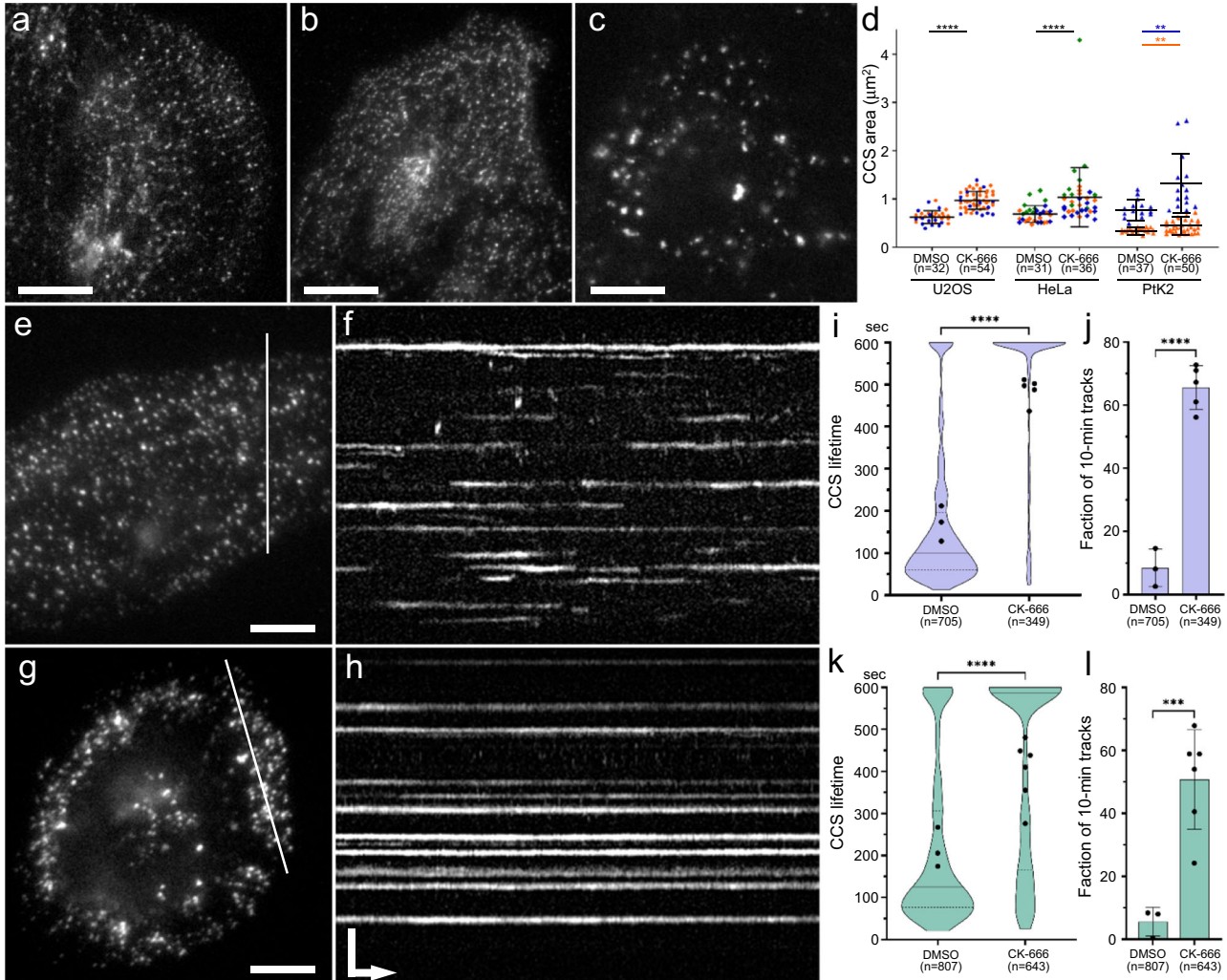

**Fig. 4 | Inhibition of the Arp2/3 complex leads to enlargement and stabilization of CCSs in intact cells. a–c** TIRF microscopy images of RFP-CLC in U2OS cells cultured in the presence of 10% FBS (**a**), or incubated overnight in a medium containing 0.1% FBS and supplemented for the last 4 h with either DMSO (**b**) or 200 μM CK-666 (**c**). **d** Quantification of the CCS area in TIRF microscopy images of the indicated cell types treated with either DMSO or 200 μM CK-666, as described for **b** and **c**. Individual experiments are color-coded. CCSs were visualized using endogenously tagged RFP-CLC (U2OS and HeLa cells) or by immunostaining of CHC (PtK2 cells; individual experiments were statistically evaluated separately due to different staining intensities). Error bars, mean ± SD; ****p < 0.0001; **p = 0.0049 (blue) or 0.0043 (orange); (unpaired two-tailed *t* test with Welch's correction); n, number of cells. **e, g** First frames of 10-min long TIRF microscopy time-lapse sequences of RFP-CLC in U2OS cells treated with either DMSO (**e**) or CK-666 (**g**) as explained for **b** and **c**. **f, h** Kymographs along lines in **e** and **g**, respectively. **i–l** Quantification of CCS dynamics showing CCS lifetimes (**i, k**) and fractions of 600-sec long traces (**j, l**) in U2OS (**i, j**) and HeLa (**k, l**) cells treated with DMSO or CK-666. Solid and dashed lines in violin plots (**i, k**) indicate median and quartiles, respectively. For CK-666-treated U2OS cells, the lines are not visible, because the median and the 75% quartile are equal to 600 s and 25% quartile is 504 s. Error bars in **j** and **l** represent mean ± SD. Black dots in **i–l** indicate average values per cell. ****p < 0.0001; ***p = 0.0005 (unpaired two-tailed *t* test with Welch's correction). Scale bars, 20 μm (**a–c**) and 10 μm (**e, g**). For both **f** and **h**, vertical bar in **h** is 5 μm, and horizontal arrow is 60 s.

along the CCS edges and partially (Fig. 5e–j, pink arrowheads; Supplementary Movie 8) or completely (Fig. 5h, red arrowheads) circumscribed them. The invasion of branched networks into CCS seams and bays became more obvious (Fig. 5e–j, yellow arrowheads). Moreover, significant fractions of CCSs became dome-shaped and spherical indicating resumption of CCS invagination, which was blocked in the absence of Arp2/3 complex activity. The newly formed invaginated CCSs were typically associated with branched actin networks abutting their perimeter (Fig. 5f–j, cyan arrowheads), thus further supporting the idea that branched actin polymerization promotes CCS invagination.

The progressive decrease in the fraction and size of flat CCSs accompanied by an increase in the fractions of invaginated CCSs (Fig. 5a, b) continued after 1.5 and 2 min of recovery (Fig. 6; Supplementary Movies 9, 10). At these time points, most actin networks

partially or completely surrounded the CCSs (Fig. 6, pink and red arrowheads, respectively) or localized between CCSs (Fig. 6, yellow arrowheads). Remarkably, invaginated CCSs at these late stages of recovery were frequently found in clusters, in which individual CCSs were separated by branched actin networks (Fig. 6e, g). These clusters most likely were formed from preexisting large flat CCSs through division and invagination promoted by branched actin networks. After 2 min washout, a significant fraction of branched actin networks formed comet tails at a side of spherical CCSs (Fig. 6, cyan arrowheads). Importantly, at all stages of recovery, branched actin networks were still localized at the edges of clathrin lattices with only minimal overhangs formed by extending filament ends.

We next monitored cell recovery after CK-666 washout in live cells by TIRF microscopy. By imaging PtK2 cells coexpressing GFP-CLC (cyan) and mCherry-cortactin (a marker of branched actin networks,

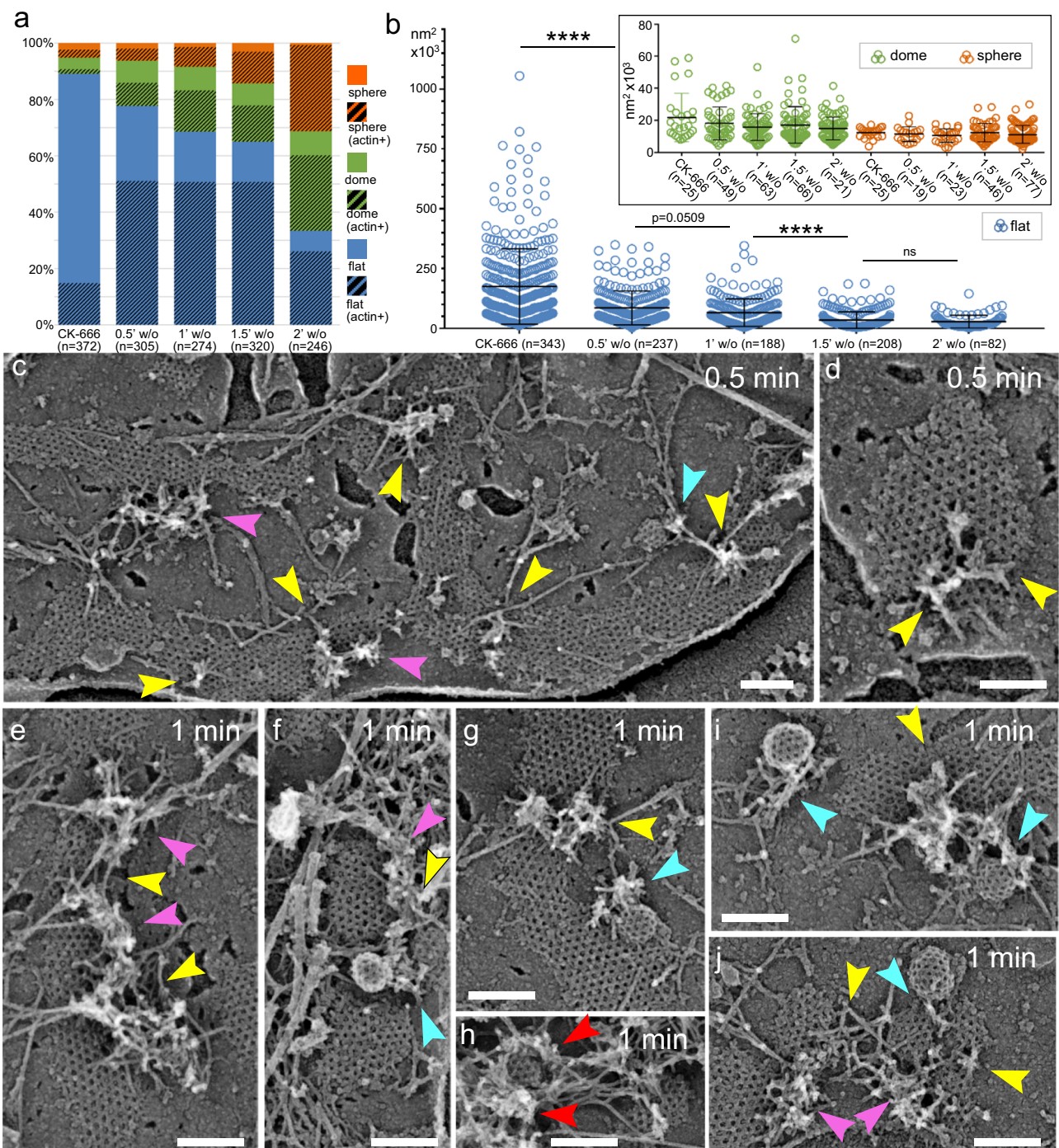

**Fig. 5 | Branched actin networks are restored after CK-666 washout and promote separation of large flat CCSs and invagination of smaller CCSs in PtK2 cells. a** Percentages of different CCS shape categories in the presence of 200 μM CK-666 and after drug washout (w/o) for indicated periods of time. Hatched colors indicate a fraction of CCSs within the given category that is associated with branched actin networks. The percentage of flat CCSs decreases over time with a concomitant increase in the percentage of dome-shaped and spherical CCSs. **b** Projection area of flat (main panel; blue) and dome-shaped and spherical (inset; green and orange, respectively) CCSs in the presence of CK-666 and after drug washout for indicated periods of time. Error bars, mean ± SD; ****p < 0.0001; ns,

p > 0.9999. Kruskal-Wallis with posthoc Dunn's multiple comparison test. Data for CK-666 in (**a**) and (**b**) are the same as in Fig. 3g and h. **c–j** PREM of PtK2 cells after washout of CK-666 for 0.5 min (**c**, **d**) or 1 min (**e–j**). At 0.5 min, small patches of branched actin networks appear at the edges of flat CCS (pink arrowheads), sometimes invading into their "bays" and "seams" (yellow arrowheads). At 1 min, branched actin networks extend along the CCS perimeter (pink arrowheads) and can completely surround them (red arrowheads), invade into bays and seams of flat CCSs (yellow arrowheads), and occasionally form comet tails at domed or spherical CCSs (cyan arrowheads), usually in immediate vicinity of flat CCSs. Scale bars, 200 nm.

red), we could directly observe division of large CCSs into smaller units, appearance of cortactin-positive patches at the clefts of the dividing CCSs, and subsequent asynchronous departure of these smaller CCSs from the TIRF field (Fig. 6i, j, arrowheads).

## Epsin knockout leads to formation of large flat CCSs

In both yeast and mammals, actin filaments are thought to transmit force to CCSs through cooperation with epsins (Ent1/2 in yeast) and Hip1R (Sla2 in yeast), as both protein families are concentrated at CCSs

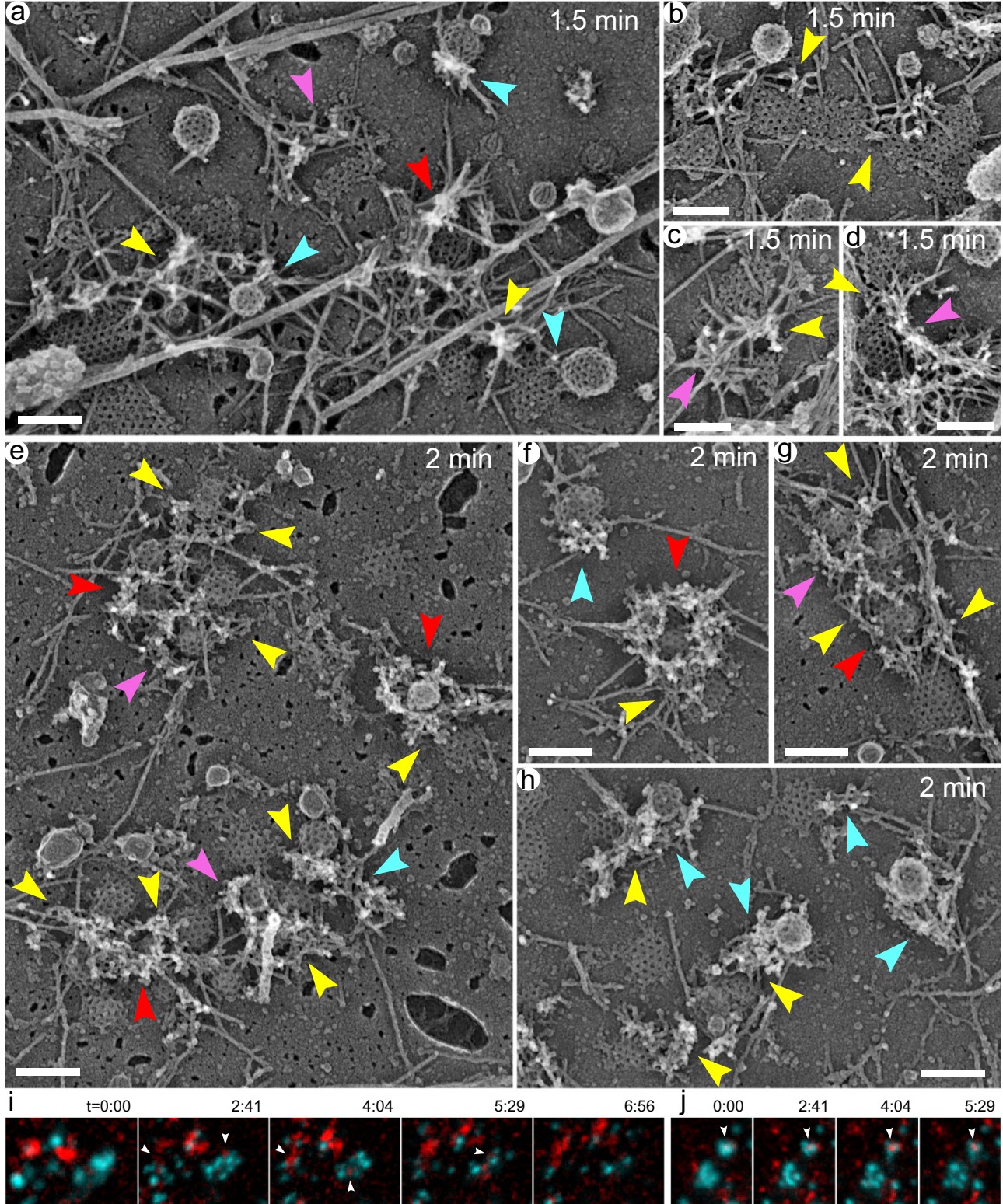

**Fig. 6 | At advanced stages of recovery, restored branched actin networks separate and surround small CCSs in PtK2 cells. a–h** PREM of PtK2 cells after washout of CK-666 for 1.5 min (**a–d**) or 2 min (**e–h**). Branched actin networks surround CCSs partially (pink arrowheads) or completely (red arrowheads) while leaving the CCS apex free of actin, localize between adjacent CCSs in CCS clusters (yellow arrowheads) and form comet tails at domed or spherical CCSs (cyan arrowheads). **i, j** Still frames from TIRF microscopy time-lapse sequences of PtK2 cells expressing GFP-CLC (cyan) and mCherry-cortactin (red) taken in the presence of CK-666 (t = 0:00) and following CK-666 washout; time after washout for individual frames is shown as min:sec. Scale bars, 200 nm (**a–h**) and 2 μm (**i, j**).

and can bind actin filaments[2], while their downregulation inhibits CME without abrogating actin polymerization[33,48]. Since recruitment of Hip1R depends on epsins[33,49], we focused here on roles of epsins in invagination and severing of flat CCSs. We used conditional epsin knockout mouse embryo fibroblasts (MEFs), which lack the epsin2 and epsin3 genes, while the floxed epsin1 gene can be deleted in the course of several days by 4-hydroxy-tamoxifen (4-OHT)-inducible Cre recombinase[33].

In the absence of 4-OHT, TIRF microscopy of epsin2/3 double knockout (DKO) MEFs transfected with GFP-CLC revealed a typical pattern of numerous small puncta that exhibited active dynamics with short lifetimes, whereas only a small fraction of CCSs lasted the entire 10-min movie (Fig. 7a, e, f, Supplementary Movie 11). Treatment of DKO MEFs with CK-666 led to a significant increase in the average CCS size and stalled CCS dynamics (Fig. 7b, e, f; Supplementary Movie 11), similar to other cell types (Fig. 4). Remarkably, the 4-OHT treatment of DKO MEFs for 5 days to produce triple knockout (TKO) MEFs closely mimicked the CK-666 treatment phenotype (Fig. 7c, e, f; Supplementary Movie 12), whereas treatment of TKO cells with CK-666 did not induce additional statistically significant changes (Fig. 7d, e, f; Supplementary Movie 12). The similar phenotypes of epsin TKO and Arp2/3 inhibition, together with a lack of CK-666 effect on TKO cells, strongly suggest that epsin and Arp2/3 complex function in the same pathway.

PREM analysis of DKO and TKO (5 days of 4-OHT treatment) MEFs demonstrated that depletion of epsins, similar to Arp2/3 complex inhibition, resulted in the formation of abundant large flat CCSs at the expense of dome-shaped and spherical CCSs (Fig. 7g–l). Consistent with the previous light microscopy data[33], branched actin networks were still abundantly present in association with CCSs in epsin TKO cells. These data suggest that in the absence of epsins branched actin networks are unable to transmit force to CCS to induce their invagination.

Given that epsins themselves are able to induce membrane curvature[50,51], it was possible that impaired CCS invagination resulted from compromised localization of epsins in the presence of CK-666. To test this possibility, we applied CK-666 or DMSO to Ptk2 cells ectopically expressing mCherry-CLC and EGFP-epsin1 and imaged cells by TIRF microscopy (Fig. 7m–q). Before CK-666 application, or before and after DMSO treatment, epsin1 formed bright puncta that strongly colocalized with CLC puncta, as well as exhibited diffuse fluorescence at the plasma membrane (Fig. 7m, n, o). However, after 2 h incubation with CK-666, we measured a significant decrease in the percentage of CLC-positive puncta colocalizing with epsin1 (Fig. 7q), as well as significant reductions in the number of epsin1 puncta (from $7.7 \pm 3.2$ to $3.6 \pm 2.3$ puncta per $100\,\mu m^2$) and their fluorescence intensity (by $77.6 \pm 7.5\%$; $p < 0.0001$; Wilcoxon matched pairs test; $n = 20$ cells from two independent experiments for both quantifications). These data show that assembly of branched actin networks promotes accumulation of epsin at CCSs.

**Microtubules can initiate actin networks growing toward CCSs**
Microtubules are known to regulate various activities of the actin cytoskeleton. We recently discovered a new aspect of such regulation—an ability of microtubule tips to initiate assembly of branched actin networks, which in turn stimulate the leading edge protrusion in neuronal growth cones[52]. Remarkably, in our unroofed PREM samples we found multiple examples of branched actin networks associated with microtubule tips (Fig. 8a–g, Supplementary Movie 13). In some cases, these microtubule-associated branched actin networks did not have obvious relation to CCSs (Fig. 8a–c) and could contribute to other functions of branched actin networks, such as membrane protrusion (e.g. Figure 8a). However, in many other cases, branched actin networks originated from microtubule tips and extended toward CCSs of various morphologies,

including spherical (Fig. 8d, e, g), dome-shaped (Fig. 8d, g) and flat (Fig. 8d–f) CCSs.

Microtubule-associated branched actin networks could even be found in CK-666-treated cells (Fig. 8h), although branched actin networks in general were scarce in these cells, and the majority of microtubule tips lacked associated actin. Upon CK-666 washout, branched actin networks gradually reappeared at the microtubule tips (Fig. 8i–l), often in configurations that are expected to push onto the edges of flat CCSs, thus promoting their division and invagination.

## Discussion
The key outstanding questions in the CME field are the mechanism of plasma membrane invagination and the source of energy driving this process. Clathrin oligomerization and actin polymerization are the main molecular machineries driving CCS invagination, although various accessory proteins also contribute[4,8,53].

Using high resolution structural analysis by PREM combined with a functional "block-and-release" approach we have advanced our understanding of the actin-dependent mechanism of CCS invagination on adherent plasma membranes in mammalian cells. First, we show that Arp2/3 complex-dependent actin polymerization is required for invagination of flat CCSs. Previously, it was thought that actin polymerization promotes CME at later CME stages, while the initial curvature is produced by other mechanisms[7,11]. Second, we uncover a previously unknown aspect of actin-assisted CCS morphogenesis – severing of large flat clathrin lattices into smaller domains that subsequently produce normally sized clathrin-coated pits and vesicles. Third, we show that branched actin networks promoting both invagination and severing of CCSs act from the CCS perimeter without obvious interaction with the apical clathrin lattice. Forth, we found that epsin family proteins are required for transmitting force from actin polymerization to CCS invagination, whereas branched actin networks in turn promote epsin accumulation at CCSs. Finally, we discovered that CCS-associated branched actin networks can be initiated from microtubule tips.

Feasibility of the flat-to-curved transition for clathrin lattices has long been a matter of debate. On one hand, high preference of clathrin for the formation of curved shapes and extensive intertwining of triskelion legs in the clathrin lattice[53] suggested that restructuring of the clathrin coat is energetically unrealistic[54]. On the other hand, experimental data support the actual existence of flat-to-curved CCS transitions in cells[12,43,55–58]. The latter idea is also supported by our current data. In contrast to prior studies which examined constitutive CME, we experimentally induced expansive fields of flat clathrin lattices and then triggered their nearly synchronous invagination. This functional approach provides strong evidence for the regulated flat-to-curved clathrin lattice transition in cells.

The confirmed ability of flat clathrin lattices to invaginate still does not answer the question how they do so. Intrinsic clathrin properties, such as a puckered shape of triskeleons and the clathrin ability to form both hexagons and pentagons, are well-suited for promoting membrane invagination[59,60], but also suggest that the flat conformation of clathrin lattices might be energetically unfavorable. The forces that flatten the clathrin lattice in cells against their preferred curvatures are not entirely clear, but likely depend on the membrane tension, cell adhesion and contribution of CLCs and other endocytic accessory proteins[53,61,62]. These forces, regardless of their nature, need to be overcome to enable productive CME and turnover of adhesive clathrin plaques.

The ability of the clathrin lattice to curve by itself when the flattening force is weakened is supported by the experiments, in which mechanical rupture of individual polygon edges led to invagination of flat CCSs[63]. Even plain incubation of unroofed cells in a buffer induced spontaneous invagination of flat CCSs[64,65], which could be accelerated by buffer acidification[57]. Apparently, incubation in a buffer caused

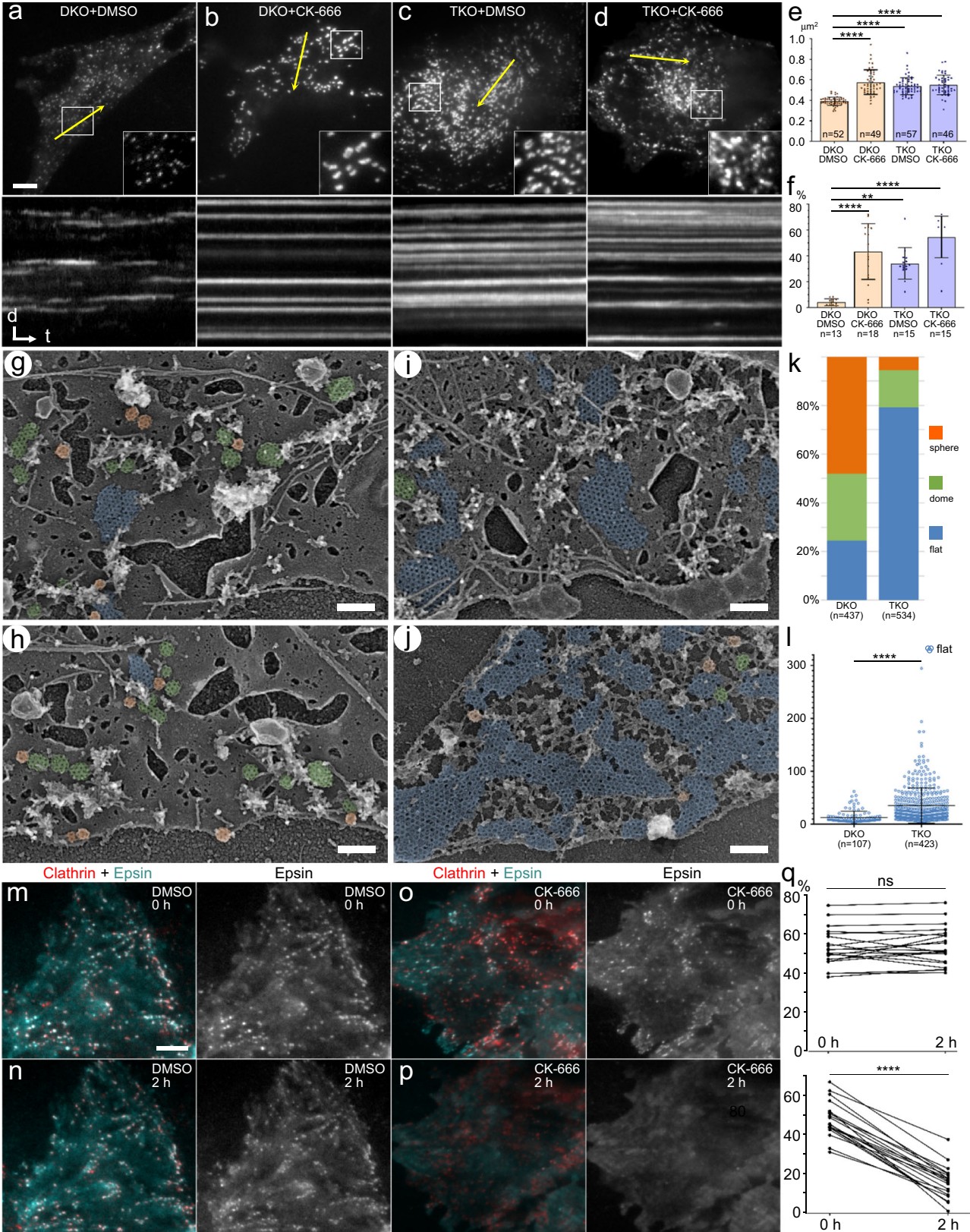

gradual deterioration of clathrin plaque adhesion, thus releasing the flattening force. Restructuring of the clathrin lattice in these conditions was likely made possible by the presence of pentagons found even in flat lattices[64,65] and incomplete saturation of lattices with clathrin molecules[66].

The fact that clathrin lattice can invaginate by itself if experimentally released from the flattening force raises the question of what mechanism naturally releases the flattening force in living cells. Our data strongly suggest that this task in mammalian cells is accomplished by the actin cytoskeleton. Previous data from budding yeast also

**Fig. 7 | Knockout of epsins mimics the phenotype of Arp2/3 complex inhibition.**
**a**–**d** (top row) First frames of TIRF microscopy time-lapse sequences of EGFP-CLC in epsin conditional DKO MEFs treated with either ethanol (**a**, **b**; labeled "DKO") or 4-OHT (**c**, **d**; labeled "TKO") for 5 days and then with either DMSO (**a**, **c**) or 100 μM CK-666 (**b**, **d**) for 4 h. Boxed regions are enlarged 2.5-fold in insets. **a**–**d** (bottom row) Kymographs taken along yellow lines in top panels. Distance (d) scale, 2 μm; time (t) scale, 1 min. **e**, **f** Quantification of TIRF microscopy data showing average CCS areas (**e**) and fractions of 600-min long tracks per cell (**f**) in indicated conditions. Error bars, mean ± SD; n – number of cells; 414 ± 285 CCSs per cell (**e**) and 156 ± 104 CCS lifetimes (**f**) per cell. Statistically significant differences are indicated. ****$p < 0.0001$; **$p = 0.0078$; Kruskal-Wallis with posthoc Dunn's multiple comparison test. **g**–**j** PREM of DKO (**g**, **h**) and TKO (**i**, **j**) MEFs. Flat (blue), dome-shaped (green) and spherical (orange) CCSs are color-coded. **k**, **l** Quantification of percentages of CCS shape categories (**k**) and the area of flat CCSs (**l**) in epsin DKO and TKO MEFs. Error bars in **l**, mean ± SD. ****$p < 0.0001$ (two-tailed Mann–Whitney test). **m**–**p** Treatment of Ptk2 cells ectopically expressing mCherry-CLC and GFP-epsin1 with DMSO (**m**, **n**) or CK-666 (**o**, **p**). Merged red/cyan images and individual epsin1 channels are shown before drug application (0 h, top row) and after treatment for 2 h (bottom row). Fluorescence intensity levels were identically adjusted for two time points in each set. **q** Percentage of CLC-positive puncta colocalizing with epsin1 puncta in the same cells before and after treatment with DMSO (top) or CK-666 (bottom); ns, $p = 0.4454$; ****$p < 0.0001$; $n = 20$ cells for each condition from 2 independent experiments; two-tailed Wilcoxon matched-pairs signed rank test. Scale bars, 10 μm ((**a**), applies to top (**a**–**d**) panels; and (**m**), applies to all (**m**–**p**) panels); 200 nm (**g**–**j**).

showed that flat CCSs invaginate only after initiation of actin assembly[12], although these findings are not without controversy[26,67]. In mammalian cells, however, pharmacological depolymerization or stabilization of actin filaments stalled CCSs in the U- or Ω-shaped configuration, suggesting that invagination per se was not inhibited[7,9–11]. Therefore, it was unclear whether actin polymerization can promote initial invagination of flat CCSs in mammalian cells.

Our data now definitively show that Arp2/3 complex-dependent branched actin polymerization promotes initial invagination of flat CCSs on adherent plasma membranes in mammalian cells. This finding does not mean that flat clathrin lattices are necessary precursors for pits and vesicles. For example, flat lattices are not detected in B16F1 cells in normal conditions. What our data suggest is that if flat CCSs do form, they are remodeled into curved ones in an actin-dependent manner. Indeed, Arp2/3 complex inhibition blocked CCS invagination, but invagination resumed after drug washout. Consistent with our data, previous studies showed that genetic or pharmacological downregulation of the Arp2/3 complex led to ~50% decrease in cargo uptake by CME[68,69]. It was important, however, to efficiently block branched actin assembly to achieve this result, which might not be done in prior studies. Under strict control of PREM, we found that at normal serum concentrations, CK-666 was less efficient, either because of partial inactivation by some serum factors or because serum agonists too strongly boosted Arp2/3 complex activation. During recovery from inhibition, flat lattices converted into spherical vesicles through intermediate formation of dome-shaped buds.

A novel aspect of the actin-promoted CCS turnover is that branched actin networks can be assembled in the vicinity of CCSs not only from unbranched actin filaments in the background[18], but also with help of microtubules. The branched actin networks growing from the microtubule tips observed in this study closely resemble analogous structures that we recently discovered in neuronal growth cones[52]. In that study, we found that the assembly of branched actin networks at the microtubule tips, and even more remarkably, throughout the growth cone, required the presence of Adenomatous Polyposis Coli (APC), a tumor suppressor protein that also functions as a microtubule plus end-tracking protein (+TIP)[70]. A mechanism underlying the formation of microtubule-associated branched actin networks likely involves an APC-dependent assembly of signaling complexes at microtubule tips, which then lead to activation of Arp2/3 complex, and consequently to a local production of pushing force for precise navigation of growth cone advance[52,70]. Our current data reveal that this type of microtubule-actin crosstalk is not unique for neuronal growth cones or for leading edge protrusion, but also functions in non-neuronal cells and for a different purpose – clathrin-mediated endocytosis – demonstrate that microtubule-associated assembly of branched actin networks has a much broader range of functions in cells than could be assumed based on a single experimental system. In the context of CME, in addition to initiating branched actin nucleation, microtubules can also provide traction for the actin network, thus directing pushing force toward the CCS. Microtubules also might regulate turnover of adhesive clathrin plaques in analogy with microtubule-dependent disassembly of focal adhesions[71].

A very interesting but still open question is how actin polymerization promotes CCS invagination. The widely held "apical pulling" model proposed for CME in yeast[12,24,25] suggests that the retrograde flow of the branched actin network drags the attached clathrin coat inward (Fig. 1a, right). Our data strongly show that this model does not apply to mammalian cells. Indeed, we find that branched actin networks barely, if at all, overlap with the clathrin coat, which is especially clear when actin organization at CCSs is viewed in 3D (see Supplementary Movies 1-3, 5, 7-10, 13). This fact conflicts with the postulated existence of strong actin-coat linkages. Furthermore, we find that barbed ends of the CCS-associated branched actin networks in mammalian cells are predominantly oriented toward the CCS, thus contrasting the apical pulling model, which predicts that barbed ends should be oriented away from the clathrin lattice[23]. Therefore, a different actin-dependent mechanism promotes CCS invagination in mammalian cells.

The actual structure of CCS-actin assemblies suggests instead an "edge pushing" mechanism in which branched actin networks push onto the perimeter of the clathrin lattice in order to promote invagination, neck constriction and vesicle departure (Fig. 8m). Interestingly, theoretical modeling has revealed that the flat-to-curved lattice transition is best explained by an increase of the line tension along the clathrin coat boundary[72], which nicely matches our data. Furthermore, branched actin polymerization has been shown to induce phase separation of lipids in model membranes[73], which perhaps can lead to line tension generation[74]. Therefore, one possibility is that branched actin networks by pushing at the membrane-clathrin lattice boundary increase the line tension at CCSs to release the flattening force and promote invagination of CCSs.

An important question is how the force generated by polymerizing actin filaments is transmitted to the clathrin lattice to induce invagination. A key role in this process, both in yeast and mammalian cells, is thought to belong to epsins (Ent1/2 in yeast) and Hip1R (Sla2 in yeast), which can bind actin filaments, CLCs and the plasma membrane, as well as each other[25,33,48]. Downregulation of either all three mammalian epsins or Hip1R alone produced similar phenotypes characterized by an impaired cargo uptake and accumulation of early U-shaped CCSs, which were more extensively associated with F-actin[33,48]. Together with similar data from yeast[25], these findings suggest that epsins and Hip proteins function together to make actin polymerization efficient in promoting CME. Given that localization of Hip1R to CCSs depends on epsin[33,49] and that Hip1R cannot bind CLC and actin simultaneously[75], we focused here on the roles of epsins. We show that, similar to Arp2/3 complex inhibition, epsin downregulation blocked invagination of flat CCSs without inhibiting branched actin networks per se. These findings support the idea that epsin-dependent force transmission from actin to CCSs is required for CME. Importantly, we show that this mechanism begins to work already for flat

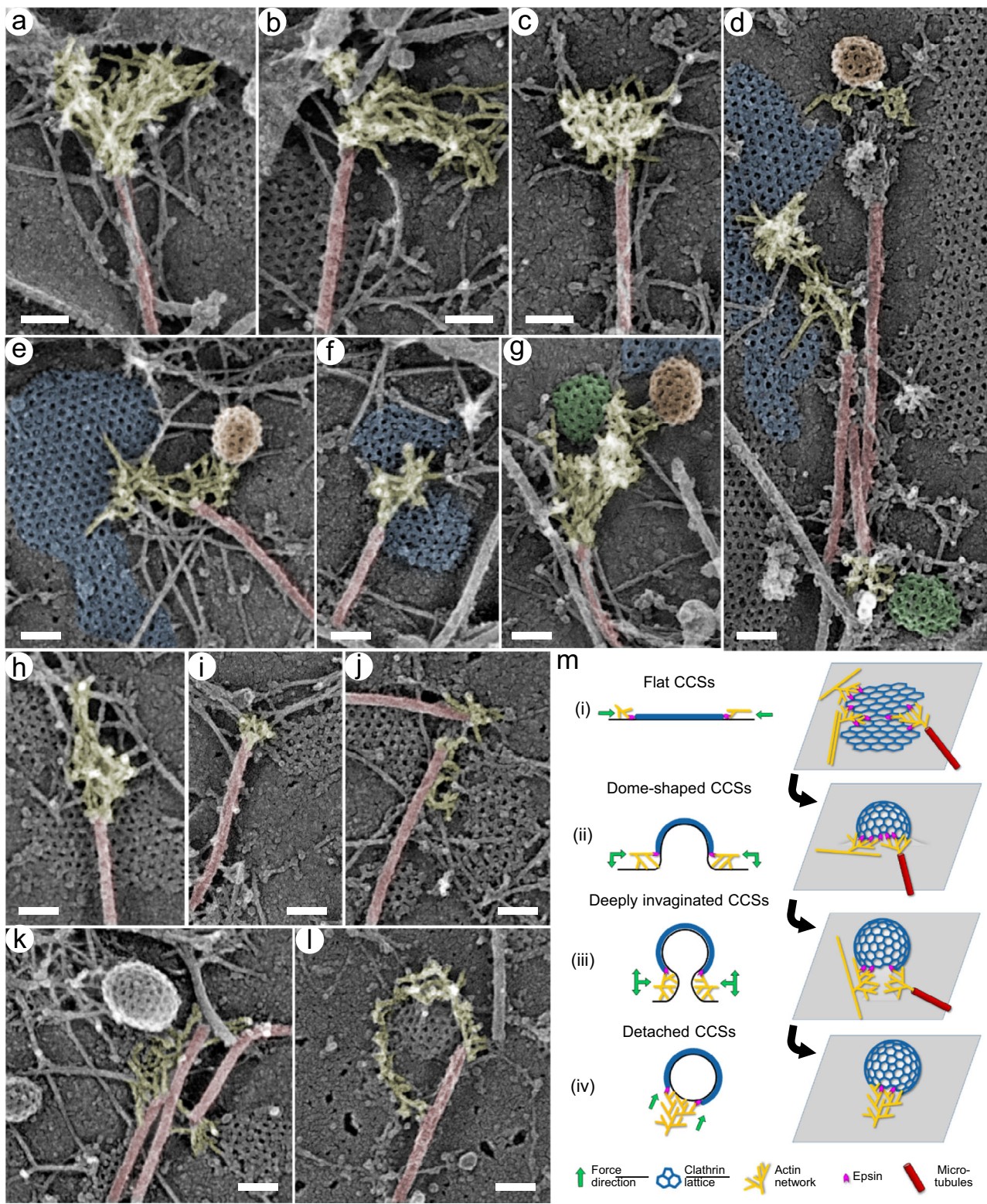

CCSs to initiate their invagination. Surprisingly, we also found that Arp2/3 complex activity is needed for efficient epsin accumulation at CCSs suggesting a feedback mechanism between actin polymerization and epsin localization than requires investigation.

According to the apical pulling model, epsin/Ent1 and Hip1R/Sla2 link actin filaments to the entire surface of CCSs, suggesting that these proteins should be distributed throughout the clathrin lattice. However, high resolution analyses in mammalian cells point to peripheral localization of both epsins[76–78] and Hip1R[79], although the data on Hip1R

localization are more controversial[44,80]. Localization of epsin to the edge of clathrin lattice is consistent with the edge pushing model, but not with the apical pulling model. The exact mechanism by which actin polymerization transmits force through epsin to induce CCS invagination remains to be fully unveiled. Given that CLCs regulate clathrin lattice dynamics and architecture[81–85], one possibility is that polymerizing actin filaments use epsins as an intermediate to mechanically induce conformational changes in CLCs and thus release an inhibitory action of CLC on clathrin lattice bending. A possibility that actin

**Fig. 8 | CCS-associated branched actin networks emerging from microtubule tips. a–g** Branched actin networks (yellow) originating from the tips of microtubules (red) in control cells cultured in the presence of 10% FBS (**a**, **d–g**) or 0.1% FBS (**b**, **c**) can extend toward the plasma membrane (**a**, **b**), to an unknown target (**c**), or toward spherical (orange), dome-shaped (green) or flat (blue) CCSs (**d–g**). Sometimes, proteinaceous material connects the microtubule tip and the branched actin network (**d**, orange CCS). **h** CK-666-treated cell. Infrequent remaining branched actin networks can still be associated with microtubule tips. **i–l** Cells after CK-666 washout for 0.5 min (**i**), 1 min (**j**), 1.5 min (**k**) and 2 min (**l**) exhibit gradual restoration of branched actin networks at microtubule tips adjacent to CCSs. Scale bars, 100 nm. **m** Edge pushing model of actin-dependent CCS morphogenesis. (i) Branched actin networks originating from microtubules or linear actin filaments

in the background separate flat CCSs into smaller domains and initiate their invagination. (ii) Branched actin networks expanding along the CCS perimeter continue to promote CCS invagination, but also constrict the CCS neck. (iii) Further expansion of branched actin networks generates subsets of pushing barbed ends that work against each other to elongate the CCSs neck, in addition to the subset that constrict the neck. (iv) Consolidation of branched actin networks into a comet tail after vesicle scission propels the newly formed vesicle away from the plasma membrane. In all stages, actin filament barbed ends push at the boundary between the plasma membrane and clathrin coat, where epsins help to transmit the force. Green arrows indicate the direction of force produced by actin barbed ends at each stage.

filaments can likewise affect epsin conformation to strengthen its localization at CCSs is also not excluded.

Large clathrin plaques can serve as cell-matrix adhesions, signaling platforms and mechanosensory devices[6,13,86,87]. All these functions require clathrin plaques to be adaptive, and therefore regulated. These regulatory mechanisms remain poorly understood. Various physical, chemical and biochemical manipulations can increase the formation of flat clathrin plaques in cells[57,88–91]. The reverse process – plaque turnover – requires actin polymerization triggered by N-WASP (activator of the Arp2/3 complex)[14], although it remained unclear how actin polymerization accomplishes this task.

We uncovered an unexpected pathway by which actin polymerization promotes turnover of large flat CCS. Surprisingly, branched actin networks appear to cut flat clathrin lattices into smaller sections, which subsequently invaginate to form normally-sized domes and vesicles. This severing function can also enable turnover of large clathrin plaques, which serve as cell adhesion sites[6,13]. When cells are recovering from Arp2/3 complex inhibition, branched actin networks appear to "burrow" into flat CCSs from their edges and are eventually found between neighboring CCSs, probably separated by these networks. These events can also be seen in live cells. The CCS severing by branched actin networks also occurs constitutively, as can be deduced from the accumulation of large flat CCSs after Arp2/3 complex inhibition and from the presence of actin networks in the bays of flat clathrin lattices, as well as between adjacent CCSs, in untreated cells. Events, in which F-actin was recruited to clefts between separating CCSs, as if pushing them apart, were previously revealed by TIRF microscopy of live cells[11]. It is interesting that actin networks tend to spread not only along the CCS perimeter, but also along the lattice seams. Perhaps, seams offer an easier access to the plasma membrane for the upstream regulators of Arp2/3 complex, such as N-WASP and its activators. Branched actin networks are able not only cut large flat lattices, but likely also prevent their formation, when they are not needed. Given that the Arp2/3 complex is regulated by numerous signaling pathways, this mechanism can control the formation and turnover of flat clathrin lattices in response to external signals.

The time course of the resumed CCS dynamics after CK666 washout suggests that severing of flat CCSs into smaller fragments is a prerequisite of clathrin lattice invagination. If we assume that the flattening force in our experimental conditions chiefly results from the clathrin lattice adhesion, which likely scales with the CCS area, then the line tension, which should be proportional to the CCS perimeter, would more efficiently release the flattening force in small CCSs with a greater perimeter/area ratio. If the flattening force mainly results from high membrane tension, the radial inward directed pushing force generated by branched actin networks can locally release membrane tension underneath the CCS and promote its invagination.

Based on our results, we propose the edge pushing model to explain a mechanism by which branched actin networks promote CME (Fig. 8m). A key point of this model is that actin filaments exert force at the boundary of the clathrin lattice while being localized outside the CCS except for a few actin filament ends that "missed the target" while

polymerizing. This edge pushing mechanism promotes severing and invagination of flat clathrin lattices (Fig. 8m, i), as we have shown here, but also supports subsequent stages of CME (Fig. 8m, ii-iv), as has been suggested by previous studies[4,11,18]. Specifically, the branched actin networks that propagate along the CCS perimeter during CCS invagination and form a full or partial "collar" around the neck of U-shaped CCSs can promote neck constriction (Fig. 8m, ii). Interestingly, a mechanical model of neck constriction suggested that asymmetric radial pressure onto the base of the neck encounters the lowest energy barrier, as compared with symmetric circumferential pressure or pressure at more apical levels of a bud[92]. These conclusions are consistent with our experimental observations that invaginating CCSs are often associated with an incomplete collar of branched actin network at their base, although nearly complete collars are also often observed. At the next stage, multiple rounds of branched actin nucleation at the neck of Ω-shaped CCSs produce barbed ends that are oriented both toward and away from the plasma membrane (Fig. 8m, iii). By working against each other, these ends elongate the neck while continue constricting it. After vesicle scission, the network consolidates into a comet tail, which drives the departure of the clathrin-coated vesicle from the plasma membrane (Fig. 8m, iv).

In conclusion, we show that Arp2/3 complex-dependent actin polymerization is able to sever large flat clathrin lattices into smaller domains and promote invagination of flat CCSs, in both cases acting from the CCS perimeter without obvious interaction with the apical clathrin lattice, while requiring epsins to function as intermediates. Strikingly, CCS-associated actin networks can be induced at microtubule tips. The ability of branched actin networks to divide and invaginate large flat CCSs has dual importance. On one hand, it can promote efficient CME in conditions of high membrane tension and/or strong cell adhesion. On the other hand, it can facilitate turnover of adhesive/signaling clathrin plaques and thereby promote cell migration and modulate cell signaling.

## Methods
### Cell culture
Genome-edited human osteosarcoma U2OS cells and human cervical cancer HeLa cells, both endogenously expressing RFP-tagged clathrin light chain A (*CLTA*) and EGFP-tagged dynamin2 (*DNM2*), were a kind gift from Dr. David Drubin, University of California-Berkeley, California, USA[41,93]. Conditional epsin DKO MEFs with deleted epsin2 and epsin3 genes and floxed epsin1 gene were a kind gift of Dr. Pietro De Camilli, Yale University, Connecticut, USA[33]. U2OS, HeLa, potoroo epithelial kidney PtK2 cells and DKO MEFs were cultured at 37 °C with 5% $CO_2$ in Dulbecco's modified Eagle's medium (DMEM) supplemented with GlutaMAX (#10569010, Thermo Fisher Scientific), 10% fetal bovine serum (FBS) (#F2442, Sigma-Aldrich) and 1% penicillin−streptomycin. Mouse melanoma B16F1 cells were cultured in DMEM/F12 (#11330-032, Thermo Fisher Scientific) supplemented with 10% FBS and 1% penicillin−streptomycin at 37 °C and 5% $CO_2$. To generate epsin TKO cells, conditional DKO MEFs were cultured in the presence of 3 µM 4-hydroxitamoxifen (4-OHT) for 2 days, transferred to a medium

containing 1 µM 4-OHT and cultured for additional 3 days. Control DKO cells were cultured in the presence of an equivalent amount of ethanol (solvent for 4-OHT).

For CK-666 treatment experiments, U2OS, HeLa and PtK2 cells were seeded on coverslips or MatTek glass-bottomed dishes, cultured overnight in DMEM with 10% FBS, transferred to DMEM with 0.1% FBS for additional ~18 h, and then to L-15 medium (#21083027, Gibco) with 0.1% FBS supplemented with either 200 µM CK-666 (#SML0006, Sigma-Aldrich) or an equivalent amount of DMSO for 4 h. For washout experiments, cells were subsequently transferred to L-15 medium with 10% FBS and incubated at 37 °C for various periods of time. For B16F1 cells, 200 µM CK-666 or equivalent amount of DMSO were applied in the presence of 10% FBS, because B16F1 cells massively detached from the substrate under low-serum conditions. By a similar reason, epsin DKO and TKO MEFs were treated with 100 µM CK-666 or equivalent amount of DMSO in the presence of 0.1% FBS for 4 hrs without the overnight preincubation in low serum.

**PREM.** For unroofing experiments, B16F1 cells were seeded on coverslips coated with laminin (25 µg/ml, #L2020, Sigma-Aldrich), and Ptk2, U2OS and Hela cells were seeded on coverslips coated with poly-D-lysine (0.2 mg/ml, #P6407, Sigma-Aldrich) or HistoGrip (1:50 dilution, #008050, Thermo Fisher Scientific)[94,95]. Epsin DKO and TKO MEFs were cultured on uncoated glass coverslips. B16F1 cells were unroofed as described previously[18]. For unroofing of other cells, coverslips with cells were quickly transferred into ice-cold PEM buffer (100 mM PIPES − KOH, pH 6.9, 1 mM MgCl₂ and 1 mM EGTA) containing 2 µM unlabeled phalloidin (#P2141, Sigma) and, in some experiments, 10 µM taxol (#T7402, Sigma-Aldrich) and unroofed by a brief (1 s) ultrasonic burst from a 1/8-inch microprobe positioned at ~45° angle ~3 mm above the coverslip and operated by Misonix XL2020 Ultrasonic Processor at 17–20% of output power. After sonication, the coverslips were immediately fixed with 2% glutaraldehyde in 0.1 M Na-cacodylate buffer, pH 7.3 for at least 20 min at room temperature.

Sample processing for PREM was performed as described previously[96,97]. In brief, glutaraldehyde-fixed cells were post-fixed by sequential treatment with 0.1% tannic acid and 0.2% uranyl acetate in water, critical-point dried, coated with platinum and carbon, and transferred onto EM grids for observation.

PREM samples were examined using JEM 1011 transmission electron microscope (JEOL USA, Peabody, MA) operated at 100 kV. Images were acquired by an ORIUS 832.10 W CCD camera driven by Gatan Digital Micrograph 1.8.4 software (Gatan, Warrendale, PA) and presented in inverted contrast. Stereo images were acquired at tilt angles ranging from −20 to +20° and presented as video animations.

### Fluorescence microscopy

For immunofluorescence staining, Ptk2 cells were quickly rinsed with PBS at 37 °C, fixed with 4% formaldehyde (Electron Microscopy Sciences) in PBS at 37 °C for 10 min, washed three times with PBS, 5 min each, permeabilized with 0.1% Triton X-100 in PBS for 10 min at room temperature, washed with PBS again, and blocked overnight with 5% BSA in PBS at 4 °C. Cells then were incubated with mouse monoclonal X-22 clathrin heavy chain antibody (#Ab2731, Abcam; 1:1000 in 5% BSA in PBS) for 45 min at room temperature. After washing five times with 0.2% BSA and 0.05% Triton X-100 in PBS, cells were incubated for 30 min with Alexa Fluor 594 goat anti-mouse IgG secondary antibody (#A11032, Thermo Fisher Scientific, 1:500 in 5% BSA in PBS) for TIRF microscopy. For STORM imaging, a secondary mouse antibody custom-labeled with Alexa Fluor 405-Alexa Fluor A647 activator/reporter dye pair[98] was used at 1:75 dilution in combination with Alexa-488 phalloidin (#8878), Cell Signaling, 1:100 in 5% BSA in PBS.

For fluorescence microscopy, the following expression constructs were used: EGFP- or mCherry-tagged clathrin light chain A (EGFP-CLC and pmCherry-CLC, respectively, gifts from J. Keen, Jefferson

University, Philadelphia, PA, USA)[37], cortactin-pmCherryC1 (Addgene, #27676, a gift from C. Merrifield, Medical Research Council Laboratory of Molecular Biology, Cambridge, UK)[39], and pEGFP-C1-Epsin1 (Addgene, #22228, a gift from P. De Camilli, Yale University, Connecticut, USA). For live-cell imaging, cells were cultured in the glass-bottomed MatTek dishes in phenol red-free L-15 medium supplemented with 10% or 0.1% FBS and maintained at 37 °C in humidified atmosphere during observation using an UNO stage-top incubator (Okolab). For CK-666 washout experiments, PtK2 cells were cotransfected with EGFP-CLC and cortactin-pmCherry and treated with CK-666 as described above. After acquiring single images at multiple positions in the presence of the drug, CK-666 was washed out using three sequential medium exchanges and time-lapse sequences were acquired. For life cell imaging of epsin DKO and TKO MEFs, cells were transfected with EGFP-CLC using Lipofectamine 3000 Reagent (Thermo Fisher) after 2 days of treatment with 3 µM 4-OHT or EtOH, cultured for additional 2 days in 1 µM 4-OHT or EtOH, replated into glass-bottomed MatTek dishes, cultured overnight in DMEM with 10% FBS and 1 µM 4-OHT or EtOH, and finally transferred to phenol red-free L-15 medium (#21083027, Gibco) for imaging. For monitoring effects of CK-666 on epsin localization, Ptk2 cells were transfected with pmCherry-CLC and pEGFP-C1-Epsin1 and treated with CK-666 as described above but for 2 h.

TIRF microscopy was performed using Eclipse TiE inverted microscope (Nikon) equipped with a CFI Apochromat TIRF 100 × 1.49 NA oil objective, iXon3 *DU-897* back-illuminated EMCCD camera (Andor) and a perfect focus system driven by Nikon (NIS)-Elements Advanced Research software (Version 4.50). The 488 nm laser and 525/50 filter were used for EGFP, the 561 nm laser and 605/70 filter were used for RFP, and the 594 nm laser and 632/50 filter were used for pmCherry. To minimize photobleaching, low laser power and exposure times and a digital gain were used.

STORM super-resolution imaging was performed in Hilo (highly inclined laminated optical sheet) illumination mode using a Nanoimager S microscope from Oxford Nanoimaging, equipped with a 100×1.4 NA oil immersion objective and Hamamatsu Orca Flash 4 V3 sCMOS camera. Images were acquired using alternating illumination with 473 nm and 647 nm laser and 15 ms exposure time. Total of 30,000 frames were recorded per image. 405 nm laser was used to induce fluorophore activation. Imaging was carried out in the OxEA buffer [50 mM β-Mercaptoethylamine hydrochloride (MEA, Sigma-Aldrich), 3% (v/v) OxyFlour™ (Oxyrase Inc., Mansfield, Ohio, U.S.A.), 20% (v/v) of sodium DL-lactate solution (L1375, Sigma-Aldrich) in PBS, pH 8–8.5 adjusted with NaOH][99]. Fluorophore localizations were determined and rendered into super-resolution images using the Nanoimager software (version 1.10.0.11025) from Oxford Nanoimaging.

### Data analyses

**Quantification of PREM images.** The shape of CCSs in most cases was obvious in 2D PREM images. In uncertain cases, the degree of CCS invagination was determined using images tilted at ±10–20 degrees. Based on the degree of invagination, CCSs were classified into three categories: (i) flat CCSs with no obvious invagination; (ii) dome-shaped CCSs that had a hemispherical or less invaginated shape with visible edges of the clathrin lattice; and (iii) spherical CCSs that had a round shape with the clathrin lattice edges not visible in 2D projection images. Dome-shaped subdomains of flat CCSs were quantified separately from the contiguous flat CCSs. The CCS area in PREM images was measured using ImageJ software.

Branched actin networks in PREM samples were identified by the presence of two or more of the following characteristics: the presence of at least several Y-shaped configurations, short filament lengths, a wide range of filament orientations, and abundance of actin filament ends. CCSs were considered to be actin-positive, if branched actin

networks were located no farther than 20 nm from the edge of the clathrin lattice. Cumulative area of actin filaments overlapping with clathrin lattice in 2D projection PREM images were measured using ImageJ and expressed as a fraction of the CCS area.

**Quantification of TIRF microscopy images.** The CCS area in TIRF microscopy images of RFP-CLC in U2OS and HeLa cells or clathrin immunostaining in PtK2 cells was quantified using ImageJ software after applying Gaussian blur (1 pixel), subtracting background with rolling ball radius of 5 pixels, automatic optimization of brightness and contrast and thresholding using Yen algorithm. Automatically selected regions of interest (ROIs) were manually corrected by separating fused ROIs if they were connected by a one-pixel bridge. Two PtK2 cells treated with CK-666, in which this procedure produced an ROI covering nearly the entire cell, were excluded from quantification. For epsin knockout MEFs, thresholding was done using Auto Local Threshold with Bernsen radius=5 and parameter 1=20 after file conversion to 8 bits without subsequent manual correction of ROIs.

For quantification of CCS dynamics, cells were imaged by TIRF time-lapse microscopy for 10 min with 2 or 3 s intervals. Kymographs were generated using the Reslice tool in ImageJ along 15 parallel lines separated by 5 μm and covering the entire field of view. After background subtraction with 5-pixel rolling ball radius, the lengths of individual traces were measured in ImageJ.

Quantification of EGFP-epsin1 colocalization with mCherry-CLC and EGFP-epsin1 intensity at mCherry-CLC-positive structures in PtK2 cells was performed using ComDet v.0.5.4 plugin for ImageJ (https://github.com/ekatrukha/ComDet) using approximate particle size of 4 pixels with options "include larger particles" and "segment larger particle" selected. The intensity threshold (in SD) was chosen for individual images to ensure proper particle selection.

**Quantification of STORM data.** The two channels of the measured STORM images were drift corrected and then aligned using fiducial markers that were included in the sample. The drift correction was performed using the built-in algorithm of the ONI microscope, and channel alignment was done using a combination of the built-in algorithm of the ONI microscope (for a coarse alignment) and a homemade MATLAB script (for a fine alignment). In this MATLAB script, 50 fiducials were automatically detected (threshold based) and fitted using a 2D Gaussian profile. The coordinates obtained for each fiducial marker in the two channels were then compared and the entire image was corrected using the average shift values of these 50 fiducial markers.

After drift correction and channel alignment, the STORM images for each channel were clustered using a Voronoi Tessellation Analysis, in which a Voronoi cluster is a collection of Voronoi polygons for which the areas are smaller than a user-defined threshold. The clusters were then visually inspected to confirm that an appropriate segmentation was obtained. The same threshold was used in all images for consistency but was independently determined for each of the channels. The threshold used for clathrin was 0.05 (chosen as such that the individual clathrin coated pits are segmented as distinct clusters) and 2.0 for actin (this maintains most of the actin localizations within a single cluster). After clustering, several rectangular regions of interest (ROI) not containing fiducial markers or stress fibers in the F-actin channel were selected in order not to bias the localization analysis (Supplementary Fig. 5a, left-most panel). In each ROI, a local area around each clathrin cluster was selected by making a polygon of the clathrin cluster and expanding it symmetrically by 1 pixel, which was equal to 117 nm (Supplementary Fig. 5a, panel 1). Only the actin and clathrin localizations in these local areas were considered for the co-localization analysis that was performed with home-made algorithms in MATLAB (see Supplementary Fig. 5a for workflow), resulting in two

numbers per ROI: the percentage of actin localizations that is co-localized with clathrin coated pits (Supplementary Fig. 5a, overlap zone in panel 3), and the percentage of non-colocalized actin localizations (Supplementary Fig. 5a, yellow shape in panel 5). To calculate densities of actin localizations within and outside of clathrin clusters in the expanded regions (Supplementary Fig. 5a, panel 1, gray zone), the clathrin cluster area was determined by making a polygon from the clathrin cluster (Supplementary Fig. 5a, panel 3, red), while the actin area outside the clathrin cluster was determined by making a polygon around the actin cluster within the expanded zone (Supplementary Fig. 5a, panel 3, cyan), calculating the joint polygon between the clathrin and actin polygons (Supplementary Fig. 5a, panel 4, green), and subtracting the clathrin polygon from this joint polygon (Supplementary Fig. 5a, panel 5, yellow). The colocalized and non-colocalized actin localization densities were determined by normalizing the total number of actin localizations within respective zones in a given ROI to the total area of these zones. This automatic procedure averages the actin localization densities among actin-positive and actin-negative clathrin clusters and therefore underestimates the difference between on-clusters versus off-clusters actin localization densities for actin-positive clathrin clusters.

**Statistics and reproducibility.** At least two independent experiments of each type have been done and produced consistent results. Specifically, the experiments shown in the following figures were repeated three times: main Fig. 1 (U2OS cells), 2 (a-m), 2 (U2OS cells in panels n and o), 4d (HeLa cells), and 7 (a-f) and Supplementary Figs. 1 and 4 (f-h, U2OS cells). All other experiments were repeated twice. Statistical analyses were performed using GraphPad Prism 9.0.0. software. Numerical data sets were first evaluated for normality by Kolmogorov-Smirnov test. For the normally distributed values, two-tailed t-test was used to compare two data sets and Welch's ANOVA test with posthoc Games-Howell test was used for multiple comparisons. For not normally distributed values, two-tailed Mann-Whitney test was used to compare two unpaired data sets, Wilcoxon test was used to compare two paired data sets, and Kruskal-Wallis with posthoc Dunn's test was used for multiple comparisons.

**Reporting summary**
Further information on research design is available in the Nature Research Reporting Summary linked to this article.

## Data availability
The quantitative data generated in this study are provided in Source Data file. All other data supporting the findings of this study are available from the corresponding author on reasonable request. Source data are provided with this paper.

## Code availability
A MATLAB script for a fine alignment of STORM images have been deposited in the GitHub database [https://github.com/melikelakadamyali/ClusterOverlap].

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

## Acknowledgements

We thank Dr. David Drubin for genetically edited U2OS and HeLa cell lines, Justin Bi for help with image quantification, Xingyuan Fang, Anil Chougule and Navish Yarna for useful comments. This work is supported by NIH grant R35 GM 140832 to TS and R01 GM 133842 to ML.

## Author contributions

C.Y. performed most experimental work and data analyses; P.C. and S.H. performed STORM imaging and analyses; M.L. supervised experiments with STORM; D.Z. helped with analysis of TIRF microscopy and PREM data; all authors contributed to preparation of figures; T.S. supervised the study and wrote the manuscript.

## Competing interests

The authors declare no competing interests.
