## [Peer Review File · Nature Communications]

Actin polymerization promotes invagination of flat clathrin-coated lattices in mammalian cells by pushing at lattice edgesEditorial Note: This manuscript has been previously reviewed at another journal that is not operating a transparent peer review scheme. This document only contains reviewer comments and rebuttal letters for versions considered at *Nature Communications*.

REVIEWER COMMENTS

Reviewer #2 (Remarks to the Author):

The authors have thoroughly revised their manuscript in light of the three reviewers' comments on the original submission to NCB. The new data, which includes quantitative STORM actin-clathrin colocalization analysis, live cell imaging of clathrin and cortactin after CK-666 washout and CK666 induced formation of flat lattices in B16 cells. Importantly, the authors found that loss of epsin phenocopies CK666 treatment. This experiment gets around possible indirect effects of CK666 on general actin dynamics as it directly links branched actin to clathrin structures. Overall, the new data strengthen the authors conclusions. The authors have also improved the discussion by openly discussing the issues raised by the three reviews including the possible limitations of PREM.

Reviewer #3 (Remarks to the Author):

Yang and coworkers have now added two key experiments to demonstrate that the Arp2/3 complex and actin play important roles to promote clathrin-coated vesicle budding from flat clathrin plaques, at stages earlier than previously demonstrated (which was basically at the stage of vesicle fission). They now directly show by TIRF live-cell imaging that large clathrin plaques generated by treatment with the Arp2/3 inhibitor CK666, resume in smaller clathrin structures upon drug retrieval, and that depletion of the epsins, which link actin to clathrin, recapitulate the effects of CK666, despite actin structures associated to clathrin still form. This is by itself an important finding, which together with their robust electron microscopy data, clearly support a model whereby branched actin filaments can promote severing and invagination of flat clathrin-coated structures in an epsin-dependent manner, probably by pushing at the clathrin-coated structure boundary, thus releasing forces opposing the intrinsic curvature of clathrin lattices.

Still though, I think the absence of actin structures associated with clathrin coats in their experimental setting is not really sufficient to rebut the actin pulling model in mammalian cells proposed by other labs. It is perfectly fine to discuss the models in the terms they do so in the answers to the reviewers, but it should not appear as a conclusion in the abstract. To do so, the authors would need to do a more careful STORM analysis including Z-reconstructions as Kaplan and co-workers did in the MBoC 2022 paper, and investigate the position of the Arp2/3 complex relative to clathrin, altogether, probably out of the scope of the manuscript, provided that they soften their conclusions in that respect.

Other points:

1. In the introduction and the respond to the reviewers, it is indicated that in yeast, it is not really demonstrated that branched actin surrounds the clathrin coat. Precisely, the sentence:

67 Electron microscopy

68 (EM) of thin-sectioned plastic-embedded yeast cells revealed a ribosome-free area around

69 CCSs, which was interpreted as an actin network”

is somehow misleading to the non-experts in the field. The ribosome free area is not really just assumed to contain branched actin filaments. This ribosome free area can be specifically labeled for actin (Mullholland et al 1994 J Cell Biol; Idrissi et al 2008 J Cell Biol) and branched F-actin binding proteins (Rodal et al 2005 MBoC; Idrissi et al 2008 J Cell Biol; Idrissi et al 2012 PNAS) at all stages of membrane invagination. Also, more recently, Abp1 (which binds Arp2/3 actin branches) has been shown to completely cover the invagination by superresolution microscopy (Mund et al 2018 Cell). The Arp2/3 complex is also particularly enriched over the clathrin coat situated at the tip of the invaginations, shown both by platinum replica electron microscopy and quantitative transmission immuno-electron microscopy (Rodal et al 2005 MBoC; Idrissi et al 2008 J Cell Biol). Even though, as the authors indicate, the precise actin filament architecture on the clathrin coats in yeast has not been elucidated, the evidence indicating the presence of a branched actin network on the clathrin coat is robust and should be referred.

2. If the authors want to include the data on the microtubules, it would seem important to discuss in more detail the previous work on the subject.

REVIEWER COMMENTS (our response is shown in blue font)

Reviewer #2 (Remarks to the Author):

The authors have thoroughly revised their manuscript in light of the three reviewers' comments on the original submission to NCB. The new data, which includes quantitative STORM actin-clathrin colocalization analysis, live cell imaging of clathrin and cortactin after CK-666 washout and CK666 induced formation of flat lattices in B16 cells. Importantly, the authors found that loss of epsin phenocopies CK666 treatment. This experiment gets around possible indirect effects of CK666 on general actin dynamics as it directly links branched actin to clathrin structures. Overall, the new data strengthen the authors conclusions. The authors have also improved the discussion by openly discussing the issues raised by the three reviews including the possible limitations of PREM.

We thank the reviewer for appreciating the additional work we have done in response to reviewers' comment. We are grateful for her/his positive evaluation of our revised manuscript.

Reviewer #3 (Remarks to the Author):

Yang and coworkers have now added two key experiments to demonstrate that the Arp2/3 complex and actin play important roles to promote clathrin-coated vesicle budding from flat clathrin plaques, at stages earlier than previously demonstrated (which was basically at the stage of vesicle fission). They now directly show by TIRF live-cell imaging that large clathrin plaques generated by treatment with the Arp2/3 inhibitor CK666, resume in smaller clathrin structures upon drug retrieval, and that depletion of the epsins, which link actin to clathrin, recapitulate the effects of CK666, despite actin structures associated to clathrin still form. This is by itself an important finding, which together with their robust electron microscopy data, clearly support a model whereby branched actin filaments can promote severing and invagination of flat clathrin-coated structures in an epsin-dependent manner, probably by pushing at the clathrin-coated structure boundary, thus releasing forces opposing the intrinsic curvature of clathrin lattices.

We thank the reviewer for appreciating the importance of new results provided in the revised version of the manuscript, robustness of our PREM analyses and accuracy of our model.

Still though, I think the absence of actin structures associated with clathrin coats in their experimental setting is not really sufficient to rebut the actin pulling model in mammalian cells proposed by other labs. It is perfectly fine to discuss the models in the terms they do so in the answers to the reviewers, but it should not appear as a conclusion in the abstract.

In response to this critique, we toned down the relevant statement in the Abstract as the following:

“This structure is hardly compatible with the widely held “apical pulling” model describing actin functions in CME.”

To do so, the authors would need to do a more careful STORM analysis including Z-reconstructions as Kaplan and co-workers did in the MBoC 2022 paper, and investigate the position of the Arp2/3 complex relative to clathrin, altogether, probably out of the scope of the manuscript, provided that they soften their conclusions in that respect.

We agree with the reviewer that these experiments are out of the scope of the current paper. First, excellent 3D STORM experiments have already been done and published by Kaplan et al. Their data clearly show that actin and clathrin always localize side-by-side in X-Y projections, which is fully consistent with our PREM data, even though Kaplan et al. avoid explicitly acknowledging this fact in words. An apparent overlap in Y-Z projections shown in this study is a result of a highly selective choice of the projection axis. Second, as for the investigating the position of the Arp2/3 complex, the presence of actin filament branches is the most story-telling and unambiguous evidence of the presence of Arp2/3 complex, as no other protein can make such branch junctions. Thus, the reviewer’s reservations are already addressed by available knowledge.

Other points:

1. In the introduction and the respond to the reviewers, it is indicated that in yeast, it is not really demonstrated that branched actin sorrounds the clathrin coat. Precisely, the sentence:

67 Electron microscopy

68 (EM) of thin-sectioned plastic-embedded yeast cells revealed a ribosome-free area around

69 CCSs, which was interpreted as an actin network”

is somehow misleading to the non-experts in the field. The ribosome free area is not really just assumed to contain branched actin filaments. This ribosome free area can be specifically labeled for actin (Mullholland et al 1994 J Cell Biol; Idrissi et al 2008 J Cell Biol) and branched F-actin binding proteins (Rodal et al 2005 MBoC; Idrissi et al 2008 J Cell Biol; Idrissi et al 2012 PNAS) at all stages of membrane invagination. Also, more recently, Abp1 (which binds Arp2/3 actin branches) has been shown to completely cover the invagination by superresolution microscopy (Mund et al 2018 Cell). The Arp2/3 complex is also particularly enriched over the clathrin coat situated at the tip of the invaginations, shown both by platinum replica electron microscopy and quantitative transmission immuno-

electron microscopy (Rodal et al 2005 MBoC; Idrissi et al 2008 J Cell Biol). Even though, as the authors indicate, the precise actin filament architecture on the clathrin coats in yeast has not been elucidated, the evidence indicating the presence of a branched actin network on the clathrin coat is robust and should be referred.

We thank the reviewer for pointing out to us the additional references that establish the presence of actin and actin-binding proteins in the ribosome-free zone visible in thin section transmission EM images around yeast's endocytic invaginations. Accordingly, we revised the above statements at the following:

"Electron microscopy (EM) of thin-sectioned plastic-embedded yeast cells revealed a ribosome-free area around CCSs, which contained actin and actin-binding proteins typically associated with branched actin networks {Idrissi, 2008; Idrissi, 2012; Kukulski, 2012; Mulholland, 1994}. However, individual actin filaments in this zone were not resolved leaving open the question of how actin filaments are organized at endocytic sites."

We decided not include the reference to Rodal et al. (2005) in this context, because their study used PREM to visualize actin, whereas we discuss thin-section EM in this statement. Also, Rodal et al. were not able to detect clathrin in their samples.

2. If the authors want to include the data on the microtubules, it would seem important to discuss in more detail the previous work on the subject.

We thank the reviewer for the suggestion to expand our discussion about the background and significance of the data about microtubules. We are much in favor of this idea and included the following text to the relevant paragraph in Discussion:

"The branched actin networks growing from the microtubule tips observed in this study closely resemble analogous structures that we recently discovered in neuronal growth cones {Efimova, 2020}. In that study, we found that the assembly of branched actin networks at the microtubule tips, and even more remarkably, throughout the growth cone, required the presence of Adenomatous Polyposis Coli (APC), a tumor suppressor protein that also functions as a microtubule plus end-tracking protein (+TIP) {Fang, 2022}. A mechanism underlying the formation of microtubule-associated branched actin networks likely involves an APC-dependent assembly of signaling complexes at microtubule tips, which then lead to activation of Arp2/3 complex, and consequently to a local production of pushing force for precise navigation of growth cone advance {Efimova, 2020; Fang, 2022}. Our current data revealing that this type of microtubule-actin crosstalk is not unique for neuronal growth cones or for leading edge protrusion, but also functions in non-neuronal cells and for a different purpose – clathrin-mediated endocytosis – demonstrates that microtubule-associated assembly of branched actin networks has a much broader range of functions in cells than could be assumed based on a single experimental system.

REVIEWERS' COMMENTS

Reviewer #3 (Remarks to the Author):

My concerns have been addressed in the revisions.